# IGSF11 is required for pericentric heterochromatin dissociation during meiotic diplotene

Bo Chen[1‡], Gengzhen Zhu[2,3‡], An Yan[1], Jing He[1], Yang Liu[4], Lin Li[5], Xuerui Yang[4], Chen Dong[2,3], Kehkooi Kee[1]*

1 Center for Stem Cell Biology and Regenerative Medicine, Department of Basic Medical Sciences, School of Medicine, Tsinghua University, Beijing, China, 2 Institute of Immunology and School of Medicine, Tsinghua University, Beijing, China, 3 Beijing Key Lab for Immunological Research on Chronic Diseases, Tsinghua University, Beijing, China, 4 MOE Key Laboratory of Bioinformatics, Center for Synthetic & Systems Biology, School of Life Sciences, Tsinghua University, Beijing, China, 5 Central Laboratory, Beijing Obstetrics and Gynecology Hospital, Capital Medical University, Beijing, China

‡ These authors contributed equally to this study as first authors.
* kkee@tsinghua.edu.cn

**Data Availability Statement:** All sequencing data files are available from the in the GEO database (GSE174752).

**Funding:** This work was supported by the Ministry of Science and Technology of China

## Abstract

Meiosis initiation and progression are regulated by both germ cells and gonadal somatic cells. However, little is known about what genes or proteins connecting somatic and germ cells are required for this regulation. Our results show that deficiency for adhesion molecule IGSF11, which is expressed in both Sertoli cells and germ cells, leads to male infertility in mice. Combining a new meiotic fluorescent reporter system with testicular cell transplantation, we demonstrated that IGSF11 is required in both somatic cells and spermatogenic cells for primary spermatocyte development. In the absence of IGSF11, spermatocytes proceed through pachytene, but the pericentric heterochromatin of nonhomologous chromosomes remains inappropriately clustered from late pachytene onward, resulting in undissolved interchromosomal interactions. Hi-C analysis reveals elevated levels of interchromosomal interactions occurring mostly at the chromosome ends. Collectively, our data elucidates that IGSF11 in somatic cells and germ cells is required for pericentric heterochromatin dissociation during diplotene in mouse primary spermatocytes.

## Author summary

For sexually reproducing species, the number of chromosomes in a mature germ cell is half that of a typical somatic cell, and its chromosome sequence is not identical to that of parental cell, these changes result from a highly specialized cell division process named meiosis. In contrast to mitosis, germ cells undergo many meiotic-specific regulatory processes during prophase I of meiosis. In mammals, the development of male and female meiotic germ cells relies on completely different microenvironment provided by sexually specialized gonadal somatic cells, but what gene is required for germ cells and gonadal somatic cells to mediate meiosis progression is largely unclear. Here, we construct a

(2018YFA0107703) to K.K.; National Natural Science Foundation of China (Grant number: 82071597) to K.K. Research in K.K. lab is partly supported by Tsinghua-Peking Center for Life Sciences. The funders played no role in the study design, data collection and analysis, decision to publish, or preparation of the manuscript.

**Competing interests:** The authors have declared that no competing interests exist.

fluorescent reporter to trace meiotic prophase in mice, and use it to examine the requirement of IGSF11 in mediating pericentric heterochromatin dissociation during meiosis.

## Introduction

Germ cells carry the task of reproduction and creating diversity for sexually reproducing species. In mammals, although both male and female germ cells undergo meiosis with the support of gonadal somatic cells, their surrounding microenvironments are quite different. In mouse ovary, female germ cells are enclosed by pregranulosa cells and enter meiosis during fetal development. In contrast, postnatal male germ cells undergo meiosis in the space between Sertoli cells within the seminiferous tubules. Although somatic cells have been shown to be required for meiotic entry and progression in mice [1–3], little is known about what proteins connecting gonadal somatic cells and germ cells are required for this process.

*Igsf11*, which belongs to the CTX family of immunoglobulin like cell adhesion molecule (IgCAMs, or immunoglobulin superfamily, IgSF), was identified in 2002 and found to be preferentially expressed in testis and brain [4]. This family of proteins are single-pass type I transmembrane glycoprotein and contain a membrane-distal V-type and a C2-type domain in their extracellular region [5]. IGSF11 functions as a cell adhesion molecule [6] and participates in regulation of synaptic plasticity [7]. Disruption of *Igsf11* or conditional knockout of *Igsf11* in Sertoli cells results in male infertility in mouse [8]. However, it is not clear if IGSF11 is required in germ cells or for progression of meiosis.

Mouse centromeres can be divided into centric and pericentric heterochromatin (PCH) domains [9]. Mouse PCH is mainly composed of major satellite repeats and enriched in repressive histone marks, such as H3K9me3 (trimethylation of histone H3 on lysine 9). During interphase and prophase I, PCH domains from different chromosomes are organized to form several chromocenters which can be cytologically visualized as bright DAPI staining chromosomal domains. During meiosis, chromosomes exhibit meiosis-specific chromosomal structures and behaviors at telomere [10], chromosome arm [11,12] and centromere [13]. In contrast to telomeres and chromosome arm regions, our understandings and characterizations of pericentric heterochromatin during meiosis is still limited.

In this study, we generated a fluorescent reporter for monitoring meiotic entry and prophase I progression. Using this system, we examined the potential role of an adhesion molecule, IGSF11, during meiosis. After analyzing the impact of *Igsf11* deletion on meiosis progression, we uncover an unexpected role of IGSF11 in regulating nonhomologous pericentric heterochromatin dissociation during male meiosis.

## Results

### Generation of mVenus-P2A-HA-Sycp3 reporter mice

To examine the potential functional requirement of *Igsf11* during meiosis, we generated a meiotic reporter system which allows us to examine the progression of meiotic prophase I and isolate meiotic cells from mouse testis.

As shown in Fig 1A, the reporter gene cassette consists of three components: a monomer fluorescent protein mVenus with higher fluorescent intensity than commonly used EGFP [14,15], a P2A linker which allows efficient *in vivo* self-cleavage of fusion proteins after translation [16], and a copy of HA epitope tag (See S1 Appendix for transgene DNA sequence). The reporter cassette was inserted right after the start codon of *Sycp3* using the CRISPR/Cas9

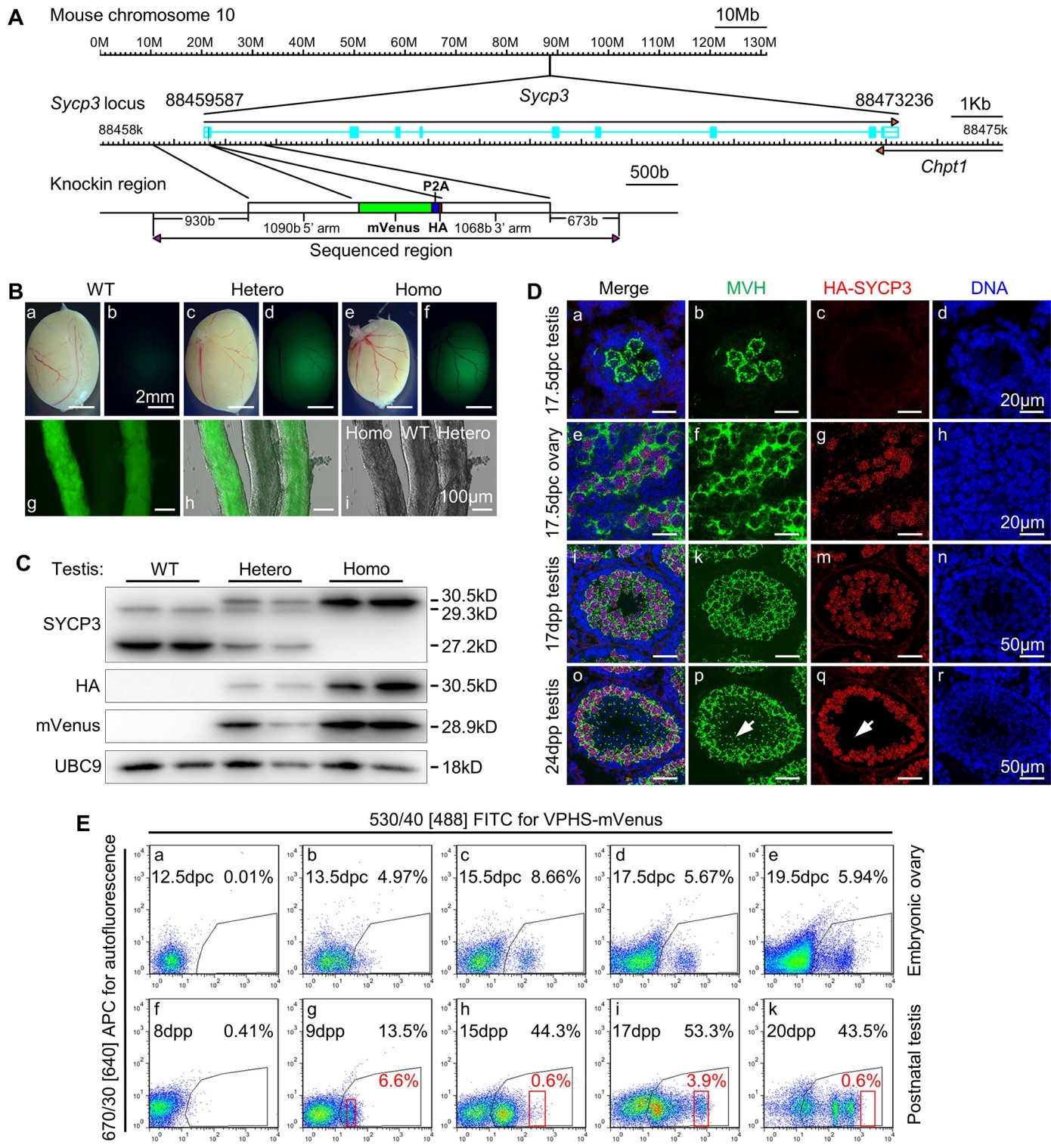

**Fig 1. Generation of mVenus-P2A-HA-Sycp3 (VPHS) meiotic reporter mice. (A)** Schematic construction of VPHS reporter. An in-frame mVenus-P2A-HA expression cassette was inserted right after the start codon of *Sycp3*. The reporter cassette is flanked by about 1kb long homologous arm at both sides in donor vector. **(B)** Detection of VPHS transgenic reporter fluorescent in 8-week-age testis. Abbreviations (the same as below): WT, wild type; Hetero, heterozygote; Homo, homozygote. **(C)** Western analysis of VPHS transgene product in adult testis with different VPHS genotypes. Target band were indicated by predicted molecular weight. The three SYCP3 bands were HA-SYCP3 (30.5kD), wild type longer SYCP3 isoform (29.3kD) and wild type shorter SYCP3 isoform (27.2kD), respectively. UBC9 served as loading control. n = 2 animals/genotype. See S1E–S1G Fig for whole Western blot results. **(D)** Localization of HA-SYCP3 in VPHS homozygote gonads by fresh-frozen immunostaining. Germ cells were localized by cytoplasmic marker MVH. Abbreviations: dpc, day postcoitum; dpp, day postpartum. **(E)**

Expression of VPHS reporter fluorescence during meiotic stages. See S2B Fig for comparison of fluorescent intensity between each time point. Fetal gonads from one pregnant mouse were pooled to detect the VPHS fluorescence for each fetal stage. Purity of the red-gating-sorted cells were further analyzed by meiotic spread respectively (S2C and S2D Fig).

"double nicking" strategy [17] by targeting at -168b and -130b upstream *Sycp3* translation start site (S1A Fig). Out of 23 targeted live offspring, 1 contained a correct knockin allele without unwanted sequence variation from +2020b to -1741b around the target site, confirmed by DNA sequencing of the amplified fragment (Fig 1A). The transgenic mice were named mVe-nus-P2A-HA-Sycp3 (VPHS) transgenic mice hereafter.

Heterozygote (Hetero) and homozygote (Homo) VPHS mice were fertile in both sexes, and generated similar litter sizes to those of wild type (WT) mice. Under the excitation light of a dissecting microscope, green fluorescence was detectable in both homozygote and heterozygote VPHS testis but not WT testis or surrounding tissues (Fig 1B), and brighter fluorescent signal was observed in seminiferous tubules of homozygotes compared to heterozygotes. Histological analysis revealed that spermatogenesis was normal in adult VPHS transgenic testis, since elongated sperms were observed in both seminiferous tubules and epididymides of homozygote or heterozygote VPHS mice, the same as wild type control (S1B Fig). However, some degenerated cells occasionally appeared in the homozygote tubules, and fewer sperms as well as more cell debris were found in the homozygote epididymis than heterozygote or wild type littermates (S1B Fig arrowheads), suggesting an overall reduced spermatogenic activity in homozygote VPHS testis, which may explain the smaller testis size for VPHS homozygote (Fig 1B).

Subsequently, the expression of *Sycp3* between different VPHS genotypes were evaluated by quantitative PCR with adult testis. Total *Sycp3* transcript levels were comparable between different genotypes (S1C Fig). As expected, we found that the expression of wild type *Sycp3* (WTS *Sycp3*) transcript decreased by half in the heterozygotes compared to wild type testes, and the expression of the knockin *Sycp3* (KIS *Sycp3*) transcript in the heterozygotes was nearly half that of the homozygotes (S1D Fig). These results suggest that there was no obvious difference in either transcription or transcript stability between wild type *Sycp3* and transgenic *Sycp3* alleles in VPHS transgenic mice.

Total proteins were also collected from adult testis with different VPHS genotypes to verify SYCP3 expression by Western blot analysis. As expected, HA-SYCP3 and mVenus were only detected in the heterozygote and homozygote VPHS samples, and their expression in homozygotes is higher than that in heterozygotes (Fig 1C). No additional larger bands were observed from blotting results using mVenus or HA antibodies (S1E and S1F Fig), suggesting the high cleavage efficiency of P2A linker under physiological conditions. Mouse *Sycp3* gene has only one transcription isoform but 2 translation isoforms (predicted to be 27.2kD and 29.3kD) [18], and the VPHS transgene was integrated right after the upstream translation start site of *Sycp3*. If both translation start site were used in the transgenic *Sycp3* transcript, there will be a 30.5kD HA tagged SYCP3 and a 27.2kD SYCP3 isoform. Interestingly, when SYCP3 polyclonal antibody was used to blot the protein extract, the 27.2kD shorter translational isoform that is the prominent band in wild type samples is absent from homozygous samples (Figs 1C and S1G). Since the total expression level of SYCP3 was not significantly different among the three VPHS genotypes (S1H Fig), we concluded that VPHS transgene insertion did not affect translational level of *Sycp3* transcript, but may have altered its translation initiation site selection.

To confirm the localization of transgenic HA-SYCP3 within meiotic germ cells, frozen sections were prepared for immunostainings using homozygote gonads (Fig 1D). HA signal was

hardly found in 17.5 dpc fetal testis that has not initiated meiosis. In the 17.5 dpc fetal ovary and the 17dpp postnatal testis, HA-SYCP3 signal showed linear patterns in the pachytene stage meiocytes that were also expressing the cytoplasmic germ cell marker MVH. Furthermore, in 24 dpp testis, no HA-SYCP3 was detected in round spermatids, which were identified by distinct nucleus morphology and a single MVH-positive chromatoid body right next to the nucleus (Fig 1D arrowheads). By comparing 17 dpp tubules of different VPHS genotypes, HA-SYCP3 staining was found to increase from heterozygote to homozygote, and colocalized well with SYCP3 staining (S1I Fig). These data suggest that transgene insertion did not influence the meiotic specific expression pattern, the localization to chromosome axes and the synaptonemal complex (SC), or the degradation after meiotic prophase of HA-SYCP3.

Mouse oocytes develop to leptotene, zygotene, pachytene and diplotene at 13.5dpc, 15.5dpc, 17.5dpc and 19.5dpc, respectively, whereas spermatocytes develop to corresponding stages at 9dpp, 15dpp, 17dpp and 19dpp, respectively. To confirm if the meiotic progress could be indicated by the fluorescence of the VPHS reporter, male and female gonads were analyzed by FACS (Fig 1E). Since the fluorescent intensity was substantially stronger in homozygote than that in heterozygote (S2A Fig), we only used homozygote VPHS mice in this analysis. In fetal ovaries, the mVenus fluorescence was hardly detected at 12.5 dpc, gradually increased from 13.5 dpc to 17.5 dpc when oocytes develop from leptotene to pachytene, and decreased at 19.5 dpc when diplotene oocytes appear (Figs 1E and S2B). In the postnatal testis, the mVenus fluorescence appeared at 9 dpp when spermatocytes enter meiosis, and increased from 15 dpp to 17 dpp. Because mVenus fluorescence decrease within spermatocyte after diplotene (20 dpp), these cells overlap with earlier stage spermatocytes in the FITC channel (Figs 1E and S2B). Furthermore, our meiotic spread staining results also confirmed the correlation between meiotic progression and mVenus fluorescent intensity in the spermatocytes (S2C and S2D Fig). Unexpectedly, mVenus fluorescent is also detected in the fetal testis from 15.5 dpc to 19.5 dpc (S2E Fig). Indeed, SYCP3 is expressed in 13.5 dpc fetal testis, but not at later fetal stages [19], and our Q-PCR result revealed that the expression pattern of *Sycp3* transcript within fetal gonads is not affected by VPHS transgene (S2F Fig), so the expression pattern of mVenus fluorescence may reflect a delayed degradation for mVenus protein within the dormant male germ cells.

## *Igsf11* is expressed by both spermatogenic cells and Sertoli cell within mouse seminiferous tubules

We generated a *Igsf11* knockout mouse by CRISPR/Cas9-mediated gene targeting in mouse embryos. The knockout allele lacks 19 base pairs in exon 2 of *Igsf11* (Figs 2A and S3B), leading to a frameshift mutation and premature translation termination. Western analysis of *Igsf11*[-/-] testis extract confirmed no expression of IGSF11(Fig 2B). *Igsf11*[+/-] male, *Igsf11*[+/-] female and *Igsf11*[-/-] female mice were fertile, but *Igsf11*[-/-] male mice were infertile. Testes of the *Igsf11*[-/-] mice were significantly smaller than testes of *Igsf11*[+/+] and *Igsf11*[+/-] (Fig 2C and 2D). Although spermatocytes exist in *Igsf11*[-/-] tubules, no elongated spermatids were found in both tubules and epididymis, leaving only round-shaped cell debris in the caput epididymis (Fig 2E).

Next, the expression pattern of *Igsf11* was analyzed by quantitative PCR. During fetal development, elevation of *Igsf11* transcript was detected in testis at 17.5 dpc, but not in littermate ovary (S3C Fig). While in postnatal testis, *Igsf11* was expressed at a low level from 1–19 dpp, the expression increased significantly at 21–23 dpp, and became even higher at 25 dpp (Fig 2F). When the expression of *Igsf11* was compared between Sertoli cells and spermatogenic cells (Figs 2G, S3D and S4C), we found a low but detectable level of *Igsf11* in Sertoli cells, which exhibited a slight upregulation after 25 dpp, and the expression level was always lower

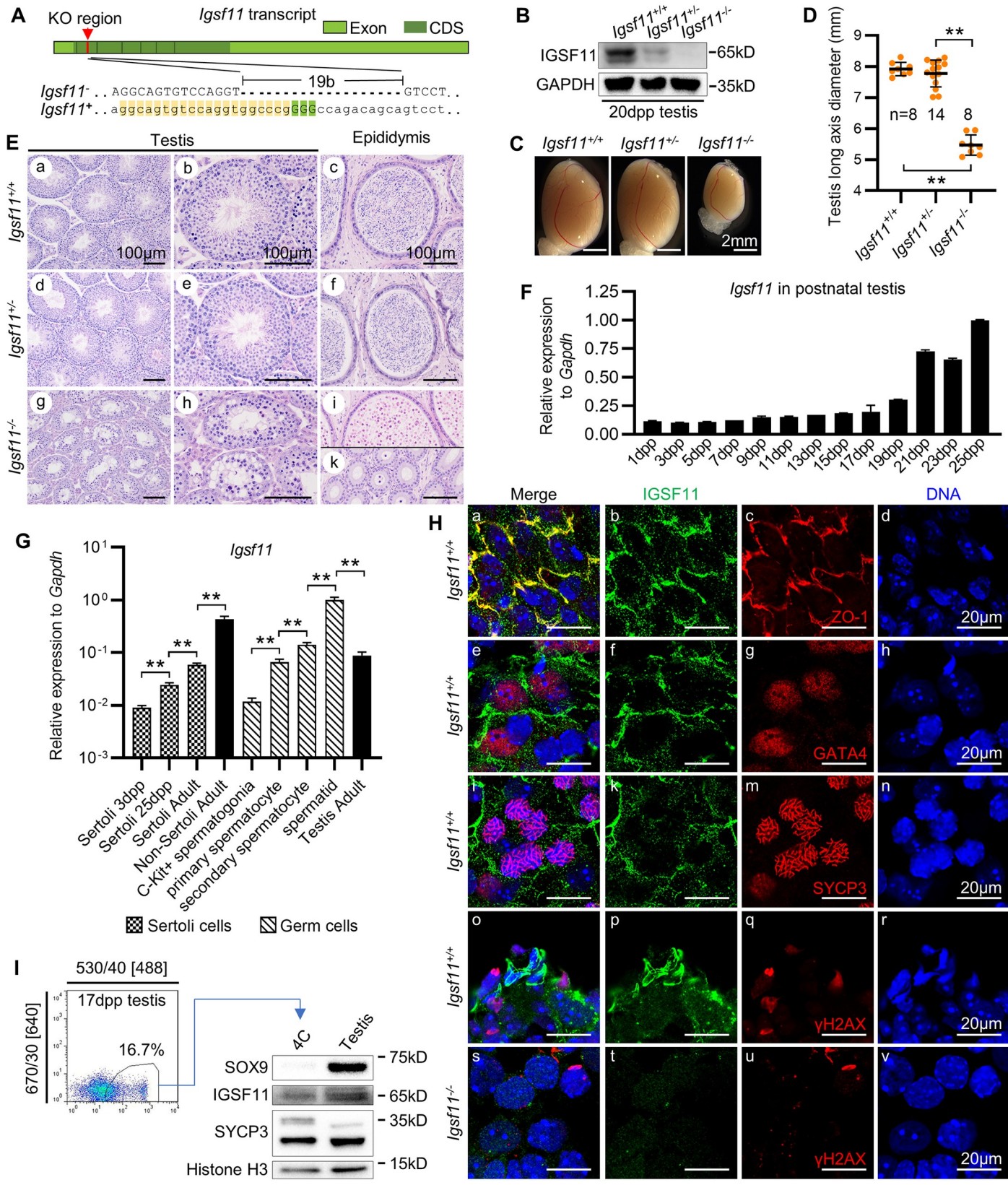

**Fig 2.** ***Igsf11*** **is expressed by both spermatogenic cells and Sertoli cell within mouse seminiferous tubules.** **(A)** Targeted knockout of *Igsf11* was achieved by Cas9-mediated 19b deletion in exon 2 of *Igsf11* in mice. Guide RNA and PAM were indicated with yellow and green background, respectively. **(B)** Western blot analysis of IGSF11 expression in 20 dpp testis with different genotypes. Note that there might be two different post-translation modification isoforms in mouse testis. GAPDH served as loading control. n = 3 animals/genotype. **(C)** Morphology of testis in adult mice of different *Igsf11* genotypes. **(D)** Diameter statistics of adult testes with different *Igsf11* genotypes. n, number of testis. **(E)** Histological analysis of adult testes from different *Igsf11* knockout genotypes. Round shaped cells were observed in the caput epididymis (i) but not in the corpora epididymis (k) in *Igsf11* knockout mice. **(F)** Expression of *Igsf11* during testis development. Graph show one of similar result from two biological repeats. Technical repeats = 2. **(G)** Expression of *Igsf11* in Sertoli cells and spermatogenic cells at different stages. Biological repeats = 3, Technical repeats = 3. See S3D or S4C Figs for Sertoli cell or spermatogenic cell purification by FACS. **(H)** Squash staining of IGSF11 in seminiferous tubules of adult mice. IGSF11 was co-stained with blood-testis barrier marker (ZO-1), Sertoli marker (GATA4), and primary spermatocyte marker (SYCP3 and γH2AX), respectively. **(I)** Western blot analysis of IGSF11 expression in primary spermatocytes sorted with the help of the VPHS reporter. SOX9 is Sertoli cell marker. Histone H3 served as loading control. n = 3 animals/genotype. Values and error bars are mean and SD. **, P < 0.01 by Student's t test.

than that in adult testis or non-Sertoli seminiferous tubule cells. *Igsf11* expression level was low in C-Kit+ differentiated spermatogonia, increased from primary spermatocyte to secondary spermatocyte, and reached the highest level in spermatid. A recent single-cell transcriptome study also showed an increasing expression pattern of *Igsf11* in spermatogenic cells from the pachytene stage onwards (S3E Fig) [20].

Next, immunostaining was performed to localize IGSF11 within squashed seminiferous tubules (Fig 2H). The specificity of our IGSF11 antibody was confirmed by Western blot and immunostaining with 293FT cells that overexpressed HA-IGSF11 (S3F and S3G Fig). First, we observed that Sertoli cell derived IGSF11 colocalizes with ZO-1, a tight-junction marker of blood-testis barrier (BTB) in the seminiferous tubules (Fig 2H, panels a-d). Besides its localization to cellular interfaces, IGSF11 also appeared as punctate aggregates (putative secretory vesicles) in cytoplasm, which was not detected in *Igsf11*-/- testicular cells (Fig 2H, panels s-v). Coimmunostaining of IGSF11 with Sertoli cell marker GATA4 (Fig 2H, panels e-h) or primary spermatocyte marker SYCP3 (Fig 2H, panels i-n) further confirmed IGSF11 expressed in the cytoplasm of both cell types. Consistent with our transcription analysis, IGSF11 expression was much higher in spermatids than in spermatocytes (Fig 2H, panels o-r). The expression of IGSF11 in primary spermatocytes was also confirmed by collecting VPHS+ primary spermatocytes (4C) cells for Western analysis (Fig 2I). In summary, mRNA and protein expression analysis both confirm that *Igsf11* is expressed in both Sertoli cells and multiple stages of spermatogenic cells.

## Adhesion molecule IGSF11 from Sertoli cells and spermatogenic cells are both required for the completion of meiosis

Cells that express IGSF11 at high level tend to form aggregates *in vitro*, indicating that IGSF11 functions as a cell adhesion molecule [6]. When ectopically expressed in human 293FT cells, IGSF11 was specifically enriched on the contact surface of two IGSF11-positive cells (S4A Fig, arrowhead), but was not enriched at the interface between expressing and non-expressing cells (S4A Fig, triangular arrowhead).

Since homophilic adhesion molecule IGSF11 is expressed in both Sertoli cells and germ cells, we next examined the effect of IGSF11 deletion in Sertoli cells or germ cells on meiotic progression by two sets of testicular cell transplantation experiments, as illustrated in Figs 3A and S4B and S1 Table. After surgical transplantation of testicular cells into the seminiferous tubules of the germ-cell depleted recipient mice, donor SSCs will colonize in the recipient testis and restore spermatogenesis. Thus, the requirement of IGSF11 in somatic cells or germ cells can be verified by the combination of donors with different *Igsf11* genotypes and recipients with different *Igsf11* genotypes, respectively. In order to evaluate the development status of the donor cells, we combined an assay utilizing the VHPS reporter system to distinguish meiotic

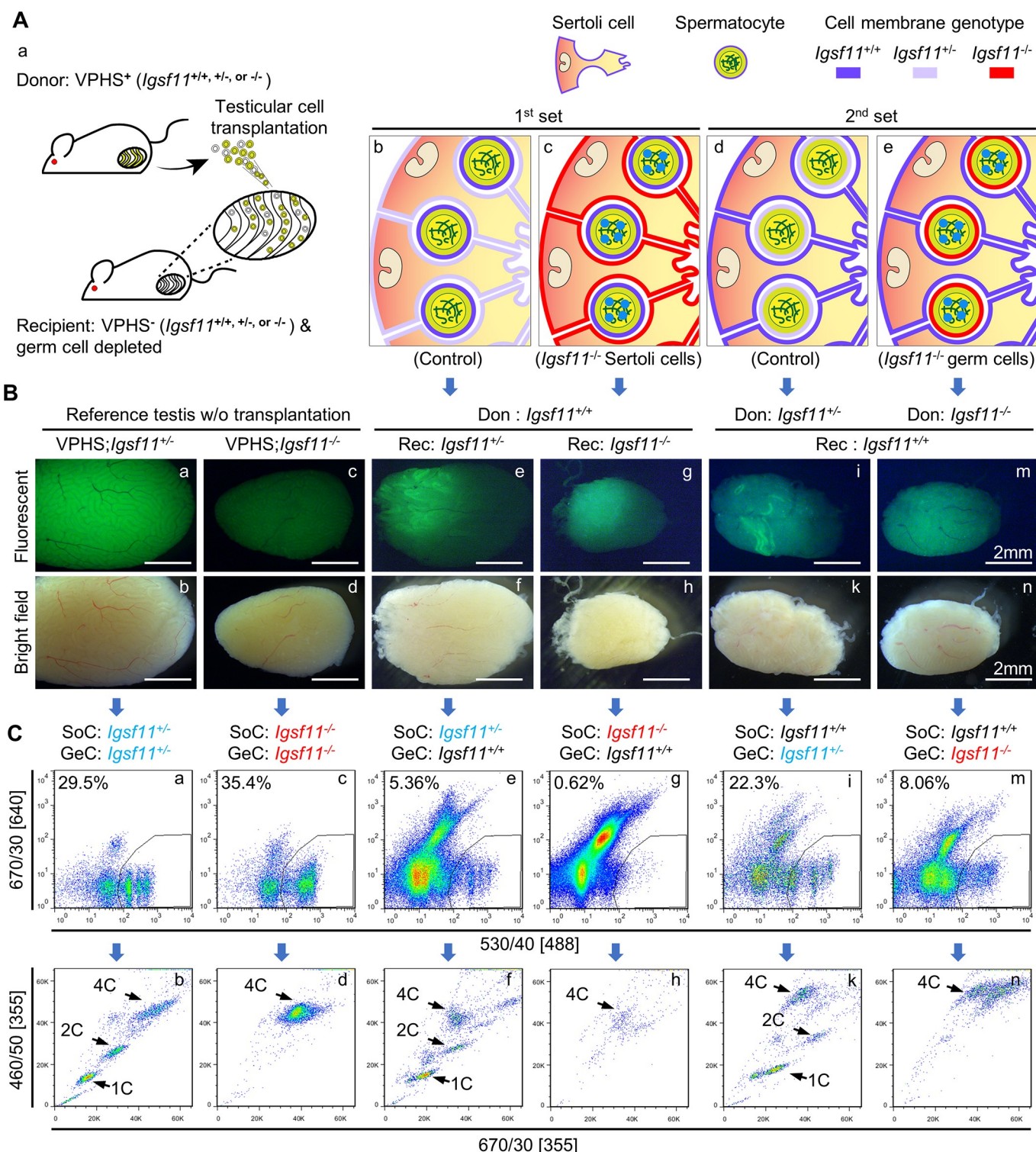

**Fig 3. IGSF11 from Sertoli cells and spermatogenic cells are both required for the completion of meiosis.** (A) Schematic views of two sets of testicular cell transplantation experiments. Genotypes of the cells were indicated by different colors of their cellular membrane: *Igsf11*$^{+/+}$, dark purple; *Igsf11*$^{+/-}$, light purple; *Igsf11*$^{-/-}$, red. In first set of transplantation, VPHS; *Igsf11*$^{+/+}$ donor testicular cells were transplanted into the testis of *Igsf11*$^{+/-}$ or *Igsf11*$^{-/-}$ recipient mice. In the second set of transplantation, VPHS; *Igsf11*$^{+/-}$ or VPHS; *Igsf11*$^{-/-}$ donor testicular cells were transplanted into the testis of *Igsf11*$^{+/+}$ recipient mice. (B) Detection of VPHS fluorescence in the recipient testes 4 month after testicular cell transplantations. (C) FACS analysis of V&H assays using recipient testes 4 month after testicular cell transplantations. Different combinations of Sertoli cells and germ cells with different *Igsf11* genotypes were indicated on the top of each lane. See S4C Fig for analysis flow; S4D Fig for gating of mVenus-positive cells; and S4E Fig for comparison between V&H assay versus traditional Hoehcst33342 staining assay. Abbreviations: Don, donor; Rec, recipient; SoC, Somatic cell; GeC, Germ cell.

cells from somatic cells and the previously established Hoechst33342 staining analysis [21] to separate 4C, 2C and 1C cells by FACS (S4C–S4E Fig) (named V&H assay for short).

As a control, we first applied these assays to the VPHS transgenic $Igsf11^{+/-}$ or $Igsf11^{-/-}$ testis. The mVenus fluorescent signal was much weaker in VPSH; $Igsf11^{-/-}$ testis than that in VPHS; $Igsf11^{+/-}$ testis (Fig 3B, panels a-d). Accordingly, V&H assay revealed that in the VPHS; $Igsf11^{+/-}$ testis, mVenus+ cells contained 4C, 2C and 1C cells (Fig 3C, panels a and b). In contrast, the mVenus+ cells in VPSH; $Igsf11^{-/-}$ testis only contained 4C cells, suggesting that the germ cells had entered meiosis and expressed HA-SYCP3, but meiosis I had not been completed (Fig 3C, panels c and d).

In the first set of experiments, we tested whether absence of IGSF11 in Sertoli cells affects meiotic progression by transplantation of VPHS; $Igsf11^{+/+}$ testicular cells into the germ-cell-depleted $Igsf11^{+/-}$ or $Igsf11^{-/-}$ recipient mice (Fig 3A, panels b and c). Four months after transplantation, when recipient testes were recovered, we found that the $Igsf11^{+/-}$ control recipient testis showed relatively strong mVenus fluorescent in the seminiferous tubules (Fig 3B, panels e and f), while only weak mVenus fluorescence could be detected in the $Igsf11^{-/-}$ recipient testis (Fig 3B, panels g and h). V&H assay revealed the presence of 4C, 2C, and 1C mVenus+ germ cells within the $Igsf11^{+/-}$ control recipient testis (Fig 3C, panels e and f), suggesting a continuous and complete meiosis from the colonized VPHS; $Igsf11^{+/+}$ donor cells. In contrast, only 4C mVenus+ cells were detected within the $Igsf11^{-/-}$ recipient testis (Fig 3C, panels g and h). This set of transplantation experiments indicated that IGSF11 in somatic cells is required for meiotic completion.

In the second set of experiments, we tested whether absence of IGSF11 in germ cells affects meiotic progression by transplanting VPHS; $Igsf11^{+/-}$ or VPHS; $Igsf11^{-/-}$ testicular cells into germ-cell-depleted wild type recipient mice (Fig 3A, panels d and e). Four months after transplantation, the mVenus fluorescent signal from VPHS; $Igsf11^{-/-}$ donor cells in wild type testis was much weaker than that from VPHS; $Igsf11^{+/-}$ donor cells (Fig 3B, panels i-n). VPHS; $Igsf11^{+/-}$ donor cells gave rise to mVenus+ cells which contained 4C, 2C, and 1C cells, suggesting completion of meiosis in the wild type recipient mice (Fig 3C, panels i and k). In contrast, only 4C mVenus+ cells were detected in the wild type recipient testes that were transplanted with VPHS; $Igsf11^{-/-}$ donor cells (Fig 3C, panels m and n), indicating incomplete meiotic progression in these cells. Hence, the two sets of transplantation experiments demonstrated that IGSF11 expression is required in both somatic cells and spermatogenic cells for meiotic completion.

## Localizations of Sertoli cells and spermatocytes in seminiferous tubules are not affected in the absence of *Igsf11*

Expression and requirement of adhesion molecule IGSF11 suggest that cellular interaction between Sertoli cells and germ cells may be important for meiosis completion, so we examined whether the absence of IGSF11 may impede cellular localization of Sertoli cells and germ cells in seminiferous tubules. Since $Igsf11^{-/-}$ testis contains mVenus+ cells arrested before meiosis I as 4C cells (Fig 3C, panels c and d), we examined cellular localization of Sertoli cells and germ cells in seminiferous tubules of 18 dpp testis before the first wave of meiosis I completion occurs and in adult testis in which some spermatocytes have completed meiosis I.

Immunostaining was performed for γH2AX, a DSB marker, and GATA4, a Sertoli cell transcriptional factor (Fig 4A, see schematic diagrams for staining results in Fig 4B, panels a-d). Within 18 dpp $Igsf11^{+/-}$ seminiferous tubules, early-leptotene (ep) spermatocytes were still close to the basal lamina, located between Sertoli cells and basal lamina, or surrounded by Sertoli cell nuclei on both sides (Fig 4A, panel a). Leptotene (le) spermatocytes migrate from the

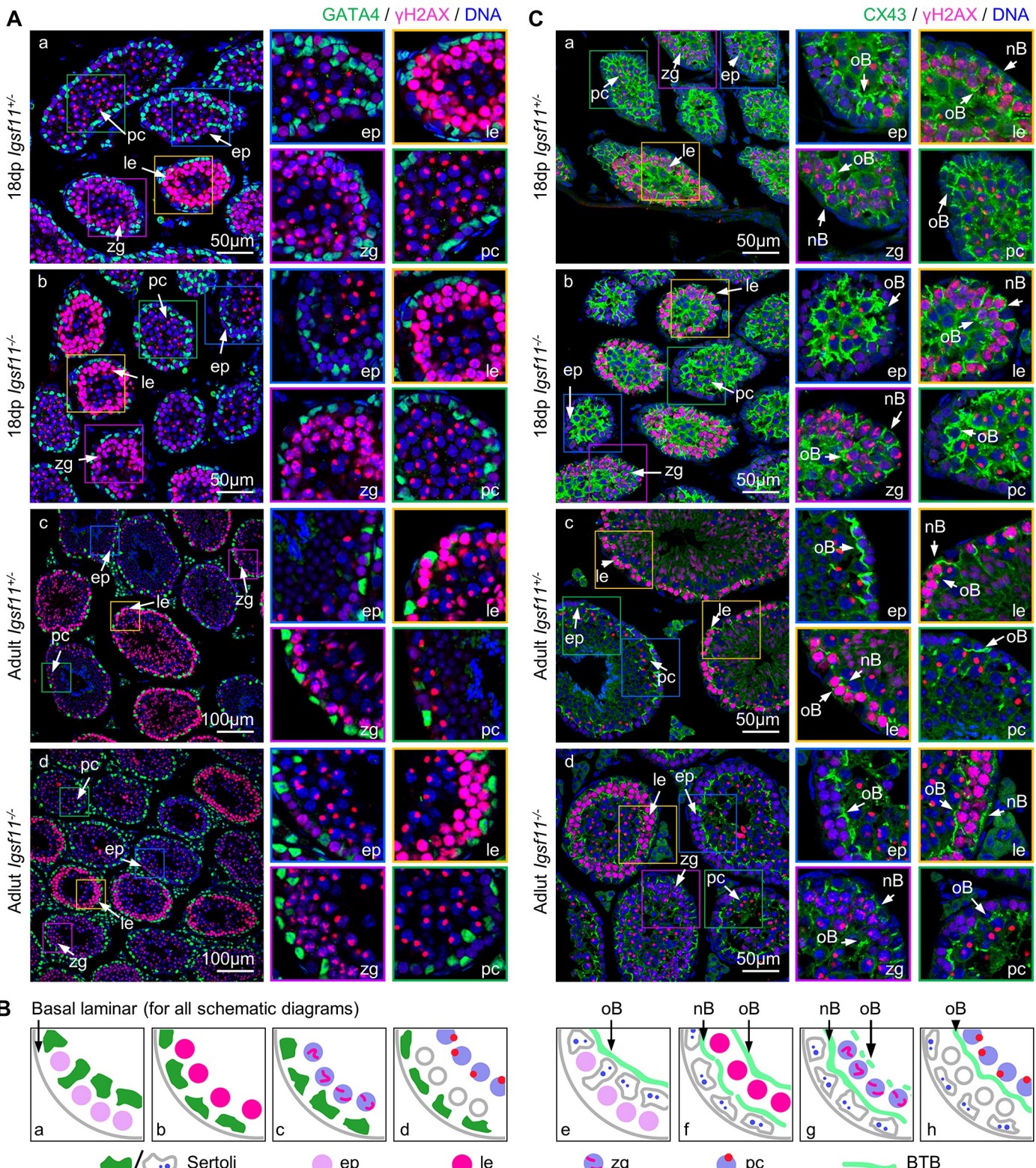

**Fig 4. Localizations of Sertoli cells and spermatocytes in seminiferous tubules are not affected in the absence of *Igsf11*.** (A) Comparison of the localization patterns of Sertoli cells and spermatocytes in the control (*Igsf11*[+/-]) and mutant (*Igsf11*[-/-]) mice using GATA4 and γH2AX as markers. (B) Schematic diagrams of the staining results in (A) and (C). (C) Comparison of spermatocytes migration and BTB reconstruction in control (*Igsf11*[+/-]) and mutant (*Igsf11*[-/-]) mice using CX43 and γH2AX as markers. Note that not all CX43 signals are restricted to BTB in the 18 dpp seminiferous tubules. Abbreviations: ep, early-leptotene; le, leptotene; zg, zygotene; pc, pachytene; nB, new BTB; oB, old BTB.

basal compartment to the adluminal compartment, whereas zygotene (zg) and pachytene (pc) spermatocytes are in the middle region of the seminiferous tubules. In 18 dpp *Igsf11*^-/- seminiferous tubules, Sertoli cells and various stage spermatocytes localized similarly to the *Igsf11*^+/- cells (Fig 4A, panel b). Furthermore, the localization pattern of Sertoli cells and germ cells was also not apparently different between adult *Igsf11*^+/- and *Igsf11*^-/- testes (Fig 4A, panel c and d). Multiple layers of leptotene spermatocytes appeared in some adult *Igsf11*^-/- seminiferous tubules, which is similar to that observed in 18dpp testis, but zygotene and pachytene spermatocytes also appeared in the same seminiferous tubules, suggesting that meiosis proceeds through pachytene. Hence, the localization pattern of germ cells at different stages in seminiferous tubules and their positions relative to Sertoli cells are not significantly affected in *Igsf11*^-/- testis.

The BTB is a highly specialized barrier formed by Sertoli cells, which divides the intratubular space into a basal compartment and an adluminal compartment. And leptotene spermatocytes migrate from the basal compartment into the adluminal compartment during the process of BTB reconstruction [22]. We next analyzed whether the migration of primary spermatocytes through the BTB is affected by the loss of IGSF11. For this purpose, the testis sections were stained for γH2AX and CX43, a gap junction protein that localizes to the BTB [23] (Fig 4C, see schematic diagrams for staining results in Fig 4B, panels e-h). In 18 dpp *Igsf11*^+/- testis, early-leptotene spermatocytes were located between the BTB and basal lamina, leptotene spermatocytes were enclosed by two layers of BTB, while pachytene spermatocytes were located in the center surrounded by BTB (Fig 4C, panel a). Similar localizations of the spermatocytes and BTB were observed in 18 dpp *Igsf11*^-/- seminiferous tubules (Fig 4C, panel b). In adult *Igsf11*^+/- seminiferous tubules, many sperm cells occupied the adluminal compartment, confining BTB to the region close to the basal lamina (Fig 4C, panel c). Although the adult *Igsf11*^-/- seminiferous tubules showed a less confined BTB, presumably because the missing post-meiotic cells no longer occupied space in the adluminal compartment, the distribution pattern of migratory spermatocytes was nonetheless similar to wild type in 18 dpp testes (two layers), and the positions of spermatocytes at different stages relative to the BTB were not affected in adult *Igsf11*^-/- seminiferous tubules (Fig 4C, panel d).

Moreover, another BTB marker ZO-1 colocalized well with CX43 in both control and mutant seminiferous tubules (S5A Fig), and no obvious defect was observed in mutant testes in transmission electron microscopy analysis of BTB before meiosis I completion (S5B Fig). These results suggest that there is no significant disruption of BTB structure in the absence of IGSF11.

## *Igsf11* deficient mouse primary spermatocytes arrest at the diplotene stage

During meiotic prophase I, programmed DSB formation and subsequent DSB repair process facilitate homolog chromosome pairing, synapsis and recombination, which is accompanied by the dynamic assembly and de-assembling of the synaptonemal complex [12]. The largely unsynapsed X and Y chromosomes are transcriptionally inactivated to form distinct chromosomal domain called XY body in spermatocytes [24]. Since *Igsf11* deficient spermatocytes may be arrested during meiosis prophase I, we co-immunostained SYCP3 with REC8, a meiosis specific component of cohesin complexes, or SYCP1, a structural component of the central element of the SC, to check whether *Igsf11* deficient spermatocytes undergo normal pairing and synapsis. Both SYCP3 and REC8 and colocalize to lateral element of the SC throughout meiotic prophase I. SYCP1 and SYCP3 start to colocalize at the homologous chromosomes undergoing synapsis since zygotene, and becoming fully colocalize at pachytene. During diplotene, homologous chromosomes start to dissociate and SYCP1 is gradually degraded from the

lateral elements [25]. Results showed that both REC8 and SYCP1 colocalize normally with SYCP3 before pachytene in the mutant spermatocytes (Figs S6A and 5A, indicating that IGSF11 is not required for homologous chromosome pairing and synapsis in mouse spermatocytes. Interestingly, *Igsf11* deficient spermatocytes also exhibited an apparently normal early/mid-diplotene SC conformation (Fig 5B) before their SC became fragmented (Figs S6A, panel h and 5A, panel h). DSB formation in the mutant spermatocytes was also confirmed to be similar to the control heterozygous spermatocytes before diplotene by co-staining for γH2AX and SYCP3 (Fig 5C). The staining results and analysis of over 2,000 mutant cells (Fig 5D and S2 Table) indicate that *Igsf11*$^{-/-}$ spermatocytes cannot develop beyond diplotene, but the appearance of earlier stage meiotic spreads was similar to the controls. These results suggested that meiotic DSB formation, DSB processing and repair, and sex body formation in *Igsf11*$^{-/-}$ spermatocytes are all carried out normally. Furthermore, meiotic entry and progression of the mutant spermatocytes were also not delayed before their developmental arrest in diplotene, as indicated by quantitative, age-specific analysis of VPHS reporter expression (S6B Fig).

Besides synapsis and DSB formation [26], germ cells with defects in a variety of different developmental pathways, including retrotransposon silencing [27,28], and meiotic sex chromosome inactivation (MSCI) [29] are also blocked by the pachytene checkpoint, which is very close to the stage that *Igsf11* mutant spermatocytes were arrested. We therefore analyzed whether *Igsf11* deficient spermatocytes were defective in these aspects. The expression levels of several retrotransposons in purified *Igsf11*$^{-/-}$ spermatocytes were comparable to that of the controls at both leptotene and pachytene stages (S6C Fig). We performed mRNA-seq using purified leptotene and pachytene stage spermatocytes, and found no differentially expressed gene (DEG) between leptotene *Igsf11*$^{-/-}$ and *Igsf11*$^{+/-}$ spermatocytes, especially for those genes that have reported function in meiosis (S6E Fig and S3 Table). 8 downregulated and 58 upregulated DEG are identified in *Igsf11*$^{-/-}$ pachytene spermatocytes. Gene ontology of 58 up-regulated genes in pachytene *Igsf11*$^{-/-}$ spermatocytes only revealed an enrichment of genes involved in aminoacyl-tRNA and amino-acid biosynthesis pathways (S6F Fig). Our mRNA-seq data also showed that from leptotene to pachytene, the expression patterns of the sex chromosome-linked genes (*Cdc25c*, *Hprt*, *Mecp2*, *Pgk1*, *Rbmy1a1* and *Ube1y1*) and compensatory genes on autosomes (*Cetn1* and *Pgk2*), are similar between *Igsf11*$^{+/-}$ and *Igsf11*$^{-/-}$ spermatocytes (S6G Fig), indicating a normal MSCI in the absence of *Igsf11*. Incorporation of histone H1t begins at mid-pachytene and increases apparently after late-pachytene in spermatocytes [30]. The incorporation of H1t started at mid-pachytene in *Igsf11*$^{-/-}$ spermatocytes as observed in the control group. Furthermore, H1t level in the arrested *Igsf11*$^{-/-}$ spermatocytes was comparable to that in diplotene stage *Igsf11*$^{+/-}$ spermatocytes (Figs 5E and S6H), suggesting that *Igsf11*$^{-/-}$ spermatocytes did develop beyond pachytene checkpoint. Interestingly, when apoptosis was analyzed by TUNEL staining, spermatocytes with various TUNEL signal intensity were observed in *Igsf11*$^{-/-}$ epididymis, but not in the adluminal compartment of the *Igsf11*$^{-/-}$ seminiferous tubules where the arrested spermatocytes were first appeared (Fig 5F and 5G), showing that apoptosis was induced in the *Igsf11*$^{-/-}$ spermatocytes after their release into the epididymis.

In summary, *Igsf11* deficient spermatocytes did not show detectable defects in synapsis, DSB formation and repair, sex body formation, retrotransposon silencing, meiotic sex chromosome inactivation, or genome-wide transcription profiles in known meiotic related genes. The meiotic entry and progression of *Igsf11*$^{-/-}$ spermatocyte was not delayed, nor were spermatocytes arrested by the pachytene checkpoint. However, *Igsf11* deficient mouse spermatocytes cannot develop beyond diplotene stage.

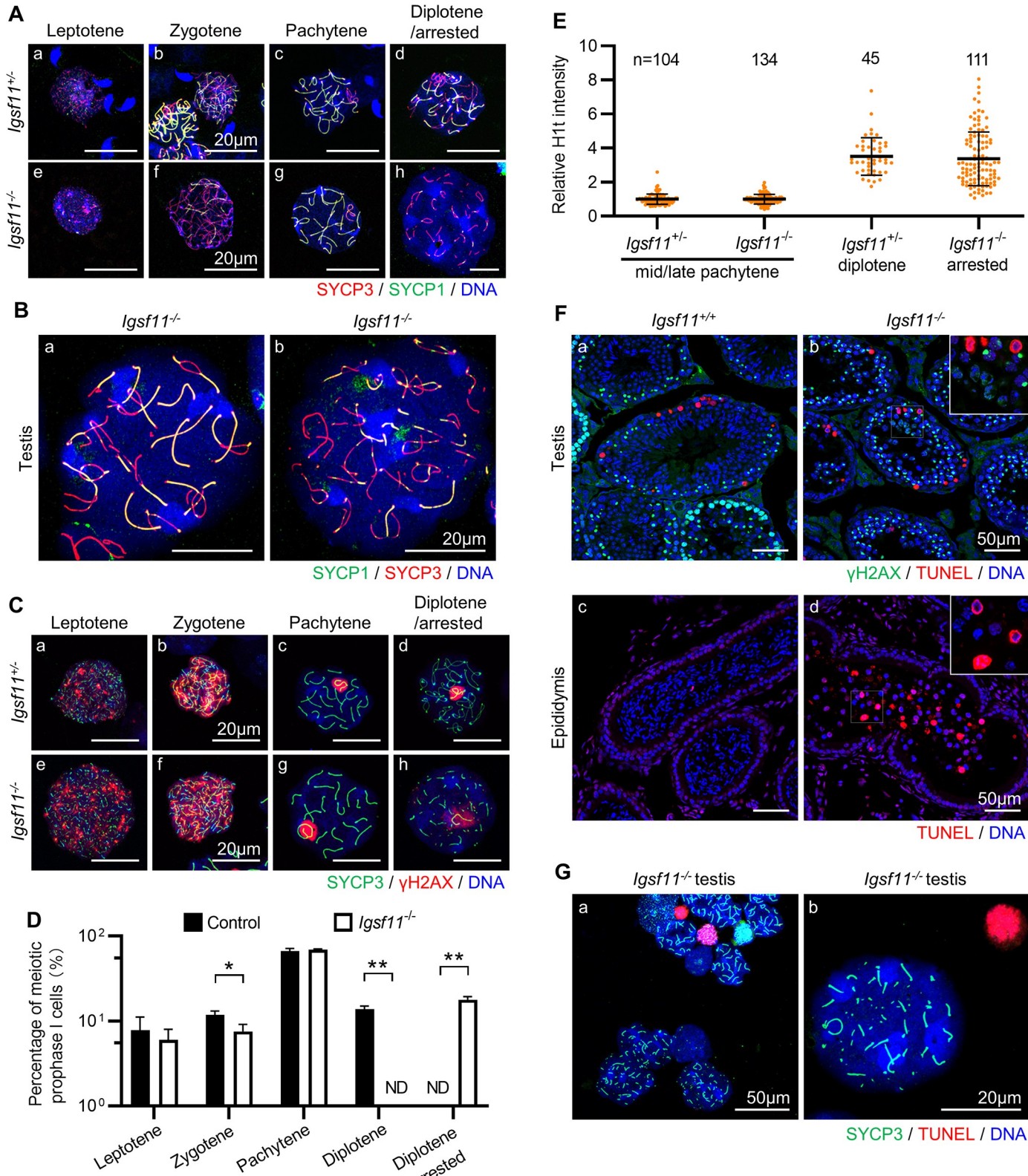

**Fig 5. Spermatocytes of *Igsf11* deficient mouse are arrested at diplotene. (A-C)** Comparison of the synaptonemal complex assembly (A and B) and DSBs repair (C) in spermatocytes of the control (*Igsf11*⁺/⁻) and mutant (*Igsf11*⁻/⁻) mice using SYCP3, SYCP1, and γH2AX as markers. Representative confocal images of diplotene stage *Igsf11* mutant spermatocytes were shown in Fig 5B. **(D)** Statistics of meiotic prophase I spermatocytes within meiotic spread samples. See S2 Table

for additional information about each biological repeat. n = 3 animals/genotype. Values and error bars are mean and SD. *, $P < 0.05$; **, $P < 0.01$ by Student's t test. **(E)** Statistics of relative H1t intensity in spermatocytes of different *Igsf11* genotypes. n, number of spermatocytes. **(F)** Apoptotic analysis using testis or epididymis sections. **(G)** Apoptosis analysis using meiotic spread specimens from *Igsf11* mutant testis. A very small number of TUNEL-positive cells, which represent a basal level of apoptosis events happened during normal spermatogenesis, were used as internal positive controls. While all the *Igsf11* mutant spermatocytes that exhibited fragmented SC complex were TUNEL-negative.

### *Igsf11* deficiency compromised dissociation of nonhomologous pericentric heterochromatin in late-pachytene mouse spermatocytes

Chromocenters are clusters of PCH from different chromosomes, and are characterized by the enrichment of H3K9me3 [9]. Compared to controls, unusually bright and large DAPI staining of chromosomes was often detected within the arrested *Igsf11*[-/-] diplotene nuclei (Figs 5A–5C and 5G and S6A and S6H), which may be putative chromocenters. Co-staining of SYCP3 and H3K9me3 revealed that PCH clusters into a few compact spherical chromocenters in early leptotene nuclei (S7A Fig) and these PCH gradually dissociate from each other from leptotene to zygotene. Complete assembly of the synaptonemal complex occurs simultaneously with the clustering of homologous and nonhomologous PCH into the same chromocenter during mid-pachytene. Despite similar assembly and dissociation patterns of PCH from leptotene to mid-pachytene in control and mutant spermatocytes, PCH patterns became significantly different starting from late-pachytene (Fig 6A–6C). Nonhomologous PCH of the control nuclei dissociated from each other since late-pachytene whereas the homologous PCH remained associated from mid-pachytene until late diplotene. In contrast, nonhomologous PCH of the mutant stayed associated to each other from mid-pachytene until late diplotene, forming 3 to 6 large chromocenters within the arrested mutant nuclei (Fig 6A–6C). Noticeably, decomposition of the SC was observed from mid-to-late diplotene in these mutant spermatocytes, which happened after PCH dissociation defect occurred (Fig 6A). In line with the staining result, proportion of spermatocytes with defect in both the SC and PCH rose and overtook proportion of spermatocytes with defect only in the PCH from 20 dpp to 26 dpp in *Igsf11* mutant (S7B Fig). To further confirm that the putative chromocenters in the arrested *Igsf11*[-/-] spermatocytes result from clustering of PCH, we co-stained the meiotic spread samples with SYCP3 and TERF1 antibodies. TERF1 binds to telomeric DNA and forms a focus on each end of a chromosome. Since all of the mouse chromosomes are acrocentric [31], one TERF1 focus (from proximal telomere) is close to the PCH for each chromosome. While the majority of the chromocenters were small and colocalized with TERF1 foci of homologous chromosomes in the control diplotene nuclei, the chromocenters in the arrested mutant nuclei were surrounded by multiple TERF1 foci which linked to the partially degenerated SC (Fig 6D, panel c, d and Fig 6E). Further co-staining of SYCP3 and anti-centromere protein antibody (ACA) revealed that all the proximal telomeres clustered to several big chromocenters in *Igsf11* mutant spermatocytes except the PCH from sex chromosome (Fig 6D, panel b arrowhead). In conclusion, absence of *Igsf11* compromised nonhomologous pericentric heterochromatin dissociation in late-pachytene mouse spermatocytes, and the PCH clustering happened before SC degeneration in the mutant nuclei.

### *Igsf11* deficiency leads to elevated interchromosomal interactions in mouse spermatocytes

As indicated by the immunostaining results that chromosomal interactions might be altered in the *Igsf11* deficient spermatocytes (Fig 6A and 6D), we further examined the arrested mutant spermatocytes by Hi-C. Toward this aim, diplotene stage spermatocytes were enriched from testes of 23 dpp *Igsf11*[+/+] and *Igsf11*[-/-] mice with the help of the VPHS reporter. Although the

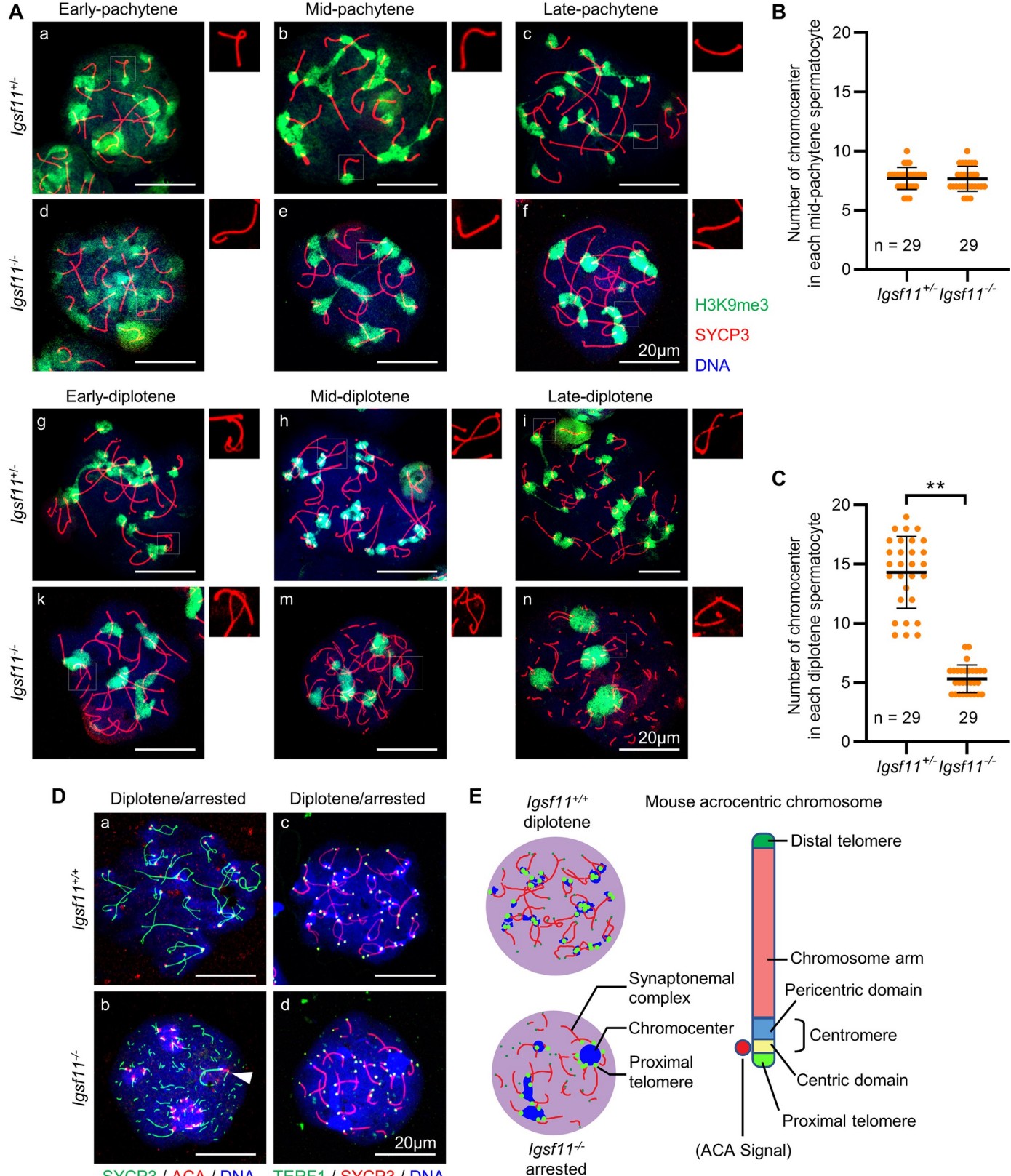

**Fig 6.** *Igsf11* **deficiency compromises nonhomologous pericentric heterochromatin dissociation in mouse primary spermatocytes. (A)** Organization dynamic analysis of pericentric heterochromatin during pachytene to diplotene in spermatocytes of the control (*Igsf11⁺ᐟ⁻*) and mutant (*Igsf11⁻ᐟ⁻*) mice using SYCP3 and

H3K9me3 as markers. The upper right corner of each image presents a partial magnification of SYCP3 staining results to distinguish the developmental stages of each spermatocyte. **(B, C)** Statistics of chromocenter number in mid-pachytene (B) and diplotene (C) spermatocytes. Values and error bars are mean and SD. **, $P < 0.01$ by Student's t test. n, number of spermatocytes. **(D)** Organization dynamic of telomeres and centromere within meiotic prophase I spermatocytes of the control and mutant mice. ACA is the abbreviation of anti-centromere protein antibody. Triangular arrowhead indicated PCH cluster of sex chromosomes. Note that SYCP3 signal was more preserved on sex chromosomes when it became fragmented from autosomes. **(E)** Schematic view of mouse acrocentric chromosome depicting the staining results shown in Fig 6D.

sorted cells contained a substantial proportion of pachytene spermatocytes, their proportions were comparable between the *Igsf11*$^{-/-}$ group and the control group (S2D Fig). First, we analyzed whether the intrachromosomal interaction is altered in the mutant nuclei. Wild type and mutant spermatocytes showed similar chromatin compartmentalization status and strength (S7C and S7D Fig). The distribution pattern of compartment A and B on each chromosome (showing chromosome 9 as example) was also similar between the wild type and mutant chromatins (Fig 7A), indicating that overall chromatin distribution is not affected in the absence of IGSF11. Furthermore, the P(s) curve analysis and intrachromosomal chromatin interaction heatmap revealed only minor differences between *Igsf11*$^{+/+}$ and *Igsf11*$^{-/-}$ spermatocytes (Fig 7B and 7C), suggesting that intrachromosomal chromatin interaction is also not affected in the absence of IGSF11.

However, we observed elevated interchromosomal interactions, especially at chromosome ends, in the *Igsf11*$^{-/-}$ spermatocytes compared to the control (Fig 7D). Although the regions with elevated interchromosomal interactions locate mainly at chromosome ends, these interactions were not limited to interactions between proximal telomere (Fig 7D, blue box), indicating that both ends of chromosomes have higher interchromosomal interactions. Therefore, we concluded that *Igsf11* deficiency leads to elevated interchromosomal interactions during meiotic diplotene.

## Discussion

By profiling the expression of *Igsf11* within seminiferous tubules and conducting testicular transplantation experiments, we showed that presence of the homophilic adhesion molecule IGSF11 is required in both Sertoli cells and spermatogenic cells for meiotic completion by mouse primary spermatocytes. Although the BTB structure is not significantly affected and mouse primary spermatocytes develop beyond the pachytene checkpoint in the absence of IGSF11, the process by which nonhomologous PCH clusters dissociate is arrested from late-pachytene onwards. IGSF11 deficiency ultimately leads to male infertility.

Although several CTX family members are expressed in mouse seminiferous tubules [32–35], IGSF11 is the only member that is essential for meiotic diplotene progression and primary spermatocyte development. Since IGSF11 is expressed in both Sertoli cells and primary spermatocytes, it has several possible nonexclusive routes of action, including mediating interactions between Sertoli cells and between primary spermatocytes independently, or mediating interactions between Sertoli cells and spermatocytes. Although it is more likely that meiotic completion by mouse spermatocytes depends on IGSF11-mediated adhesion between Sertoli cells and spermatogenic cells, our current experimental evidence has not ruled out other possible routes. Future molecular and cellular experiments are needed to decipher the molecular and cellular mechanisms of IGSF11 during spermatogenesis.

Spermatocytes that have defects in meiosis-related developmental processes such as DSB formation and DNA repair are mainly eliminated through apoptosis during the pachytene stage [36,37]. In the absence of IGSF11, mouse primary spermatocytes were instead arrested at diplotene without apparent defects at earlier stages. The novel spermatocyte elimination stage that we observe in *Igsf11* mutants suggests the existence of a previously unknown surveillance

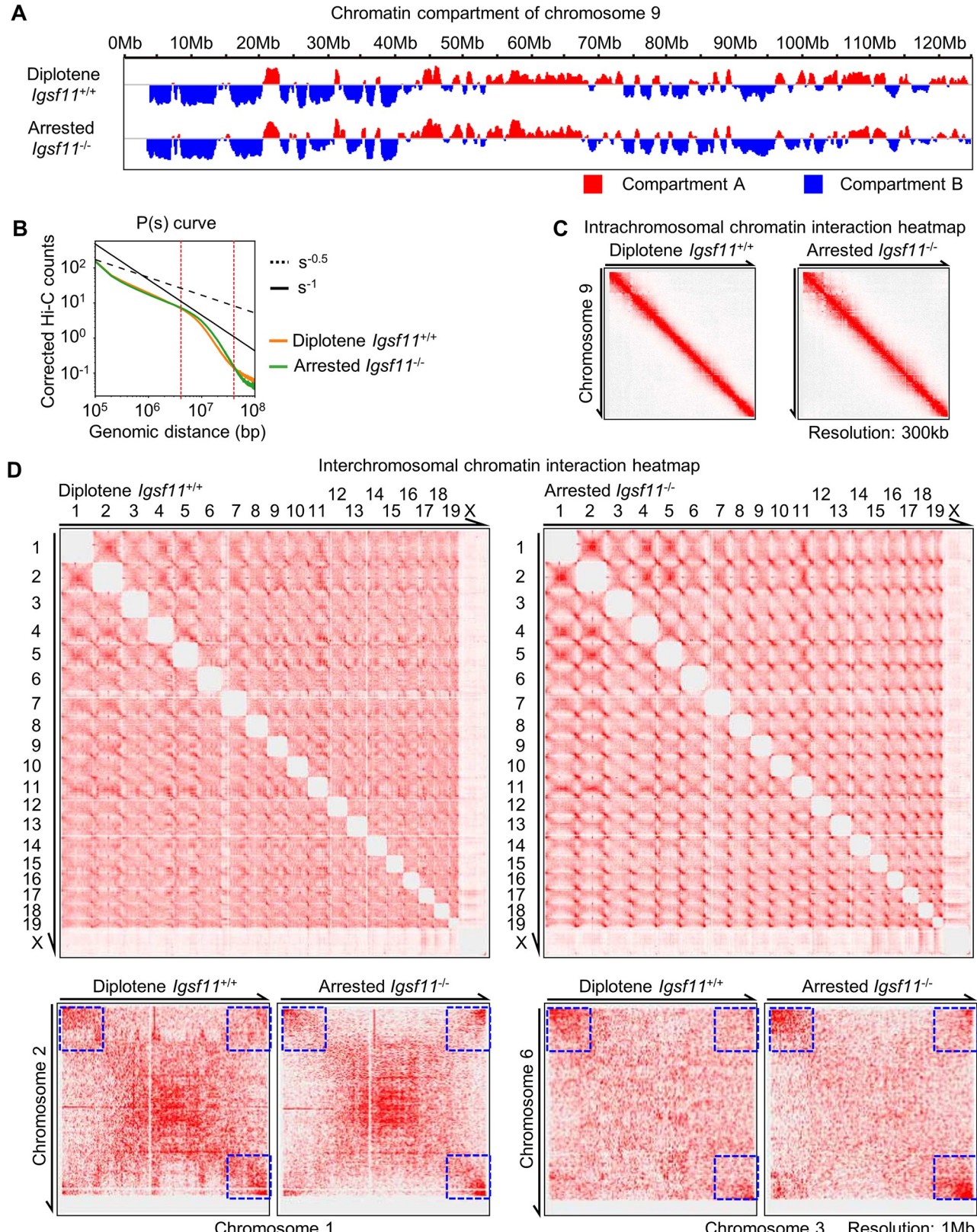

**Fig 7. *Igsf11* deficiency leads to elevated interchromosomal chromatin interactions in diplotene spermatocytes. (A)** Chromatin compartment analysis (chromosome 9 as example) represented by the PC1 values for Hi-C data of spermatocytes of the control (*Igsf11*⁺/⁺) and mutant (*Igsf11*⁻/⁻)

mice. Red represents compartment A and blue represents compartment B. Biological repeats = 2. **(B)** The corrected Hi-C counts as a function of genomic distance (P(s) analysis) (averaged across all chromosomes) for spermatocytes of the control and mutant mice. **(C-D)** Intrachromosomal (C) and interchromosomal (D) interaction heatmaps for spermatocytes of the control and mutant mice. Blue dotted boxes highlight the regions showing higher level of interchromosomal interactions at the mutant chromosomes than the control chromosomes.

mechanism and a novel checkpoint that works after the pachytene checkpoint. This putative checkpoint may be responsible for surveying defects during or after pachytene such as failure of nonhomologous PCH clusters to dissociate. Since IGSF11 is a typical membrane protein that may serve as a starting point of the potential signal transduction pathway, downstream factor(s) of IGSF11 in cytoplasm may be needed in spermatocytes to mediate this putative checkpoint.

The organization of PCH in cells varies significantly in different cellular states. Within mouse spermatocytes, PCH undergoes dynamic assembly and dissociation during meiotic prophase I, and more importantly, the dynamics of PCH is highly correlated with that of the SC (Figs 6A and S7A, and [38,39]), suggesting that the dynamics of PCH is also regulated and coordinated by the meiotic program. A few proteins have been reported to be involved in the regulation of PCH in mouse spermatocytes, including SUV39H1/SUV39H2 [40], YY1 [41], H1Pγ [42], and DICER [43], but the functions of none of these proteins is restricted only to the PCH region. It is therefore important to note that the phenotypes observed in these mutants may also be the result of a deficiency of protein functions in other chromosomal or cellular regions. Interestingly, in the absence of IGSF11, elevated clustering in the PCH region occurs prior to SC degeneration at the chromosome arm, indicating that IGSF11 is more likely to restrict its regulation on PCH but not SC and other chromosomal events during meiosis prophase I.

Meiotic prophase I is one of the most complicated developmental stage during germ cell development, yet we still lack reported system that allows us to dissect each stage of prophase in male and female meiocytes. Although several germ cell reporter mice have been generated (Summarized in S4 Table), none of these reporters could precisely indicate meiotic entry and prophase I progression. In this study, we generated a SYCP3 reporter system for monitoring meiotic entry and progression of meiotic prophase I. Combining the VPHS reporter with Hoechst33342 staining analysis, the V&H assay enable precise analysis of the developing meiotic cells, even when the percentage of isolated germ cells is low (S4E Fig). Therefore, we believe the VPHS reporter established in this study will provide a reliable tool to study meiotic prophase in greater details, such as chromosomal dynamics in both male and female meiocytes.

## Materials and methods

### Ethics statement

All animal experiments and procedures are approved by Animal Ethics Committee of Tsinghua University (approval no. 17-JJK1).

### Mice and welfare statement

*Igsf11*^−/− mice were crossed to VPHS transgenic reporter mice for V&H assay. See S5 Table for detailed information about genotyping of VPHS and *Igsf11*^−/− mice. Amh-*Cre* mice [44] were mated to *mT/mG* mice, a double-fluorescent Cre reporter line [45], for purification of Sertoli cells by FACS (S3D Fig). All mice were maintained in C57BL/6 background and housed under controlled lighting conditions (12L:12D).

### Guide RNA design *and in vitro* transcription

Guide RNAs were designed by an online software [46], and inserted into pX335 plasmid (addgene ID: 42335). T7 promoter was added upstream the expression cassette of Cas9 D10A nickase and guide RNA, respectively, through polymerase chain reaction (PCR) from the pX335 plasmid. The PCR products were purified by DNA Clean & Concentrator (ZYMO, D4033) and used as *in vitro* transcription template. Capped Cas9 mRNA and uncapped sgRNA were synthesized by an *in vitro* transcription kit (New England Biolabs, E2040) according to the manufacturer's protocol. After removal of the template by DNase I (New England Biolabs, m0303), RNAs were cleaned up by RNeasy Mini kit (Qiagen, 74104) and dissolved in RNase-free water.

### Generation of transgenic mice

To confirm the target efficiency of our system, 20ng/μL *in vitro*-transcripted Cas9 D10A nickase mRNA and 10ng/μL guide RNAs were microinjected into the pronuclear mouse embryos with Eppendorf TransferMan NK2 microinjection system, and DNA fragment flanking the target site was amplified when embryos developed to blastocysts. Five different indels that caused by Cas9 induced double strands breaks were detected from twenty sequenced fragment clones (S1A Fig).

In order to generate the transgenic mice, pronuclear stage C57BL/6 mouse embryos were injected with targeting mix (2ng/μL donor DNA, 20ng/μL Cas9 mRNA, and 10ng/μL of each sgRNA), and cultured overnight in KSOM medium. The next day, 2-cell stage embryos were surgically transferred into oviduct of pseudopregnant mice.

### Living imaging

Testes from different genotypes were examined under a fluorescent dissecting microscope (Leica, M165Fc) to detect mVenus signal. Then, seminiferous tubules were placed under an inverted fluorescent microscope (Nikon, EclipseTi) for imaging.

### Quantitative real-time polymerase chain reaction

One whole testis (8-week-old) was first stabilized in RNAlater (Qiagen, 76104), and homogenated in 1.5mL the buffer RLT. Then, 350μL lysate was taken and progressed to RNA purification using RNeasy Mini kit (Qiagen, 74104). 1ug of each total RNA was used for revers transcription by PrimeScript RT reagent Kit with gDNA Eraser (TaKaRa RR047A). Primers specific for each gene were designed by Primer Blast and primers for retrotransposons were described previously [47]. See S6 Table for primer sequences. Quantitative RT-PCR was performed with iTaq Universal SYBR Green supermix (Bio-Rad, 172–5122).

### Western blot

Mice testes from different genotypes were lysed in RIPA buffer (CoWin Biosciences, CW2333). Protein lysates were separated by 8%-12% SDS-PAGE gel electrophoresis, electro-transferred onto a nitrocellulose membrane, and processed according to standard procedures. Antibodies were diluted in TBST containing 5% non-fat milk (BD Biosciences, 232100). First antibodies were incubated overnight at 4˚C, and secondary antibodies were incubated at room temperature for 1 hour. There times washing by TBST were performed after each incubation. See S7 Table for antibody information. After adding HRP substrate (Millipore, WBKLS0100), the chemiluminescent signals were detected by an imaging system (Bio-Rad, ChemiDoc XRS$^+$).

## Hematoxylin-eosin staining

Testes and epididymides from 8-week-old mice with different genotypes were fixed in Bouin's solution overnight at 4°C. Then, samples were imbedded in paraffin, sectioned at 6μm, and subjected to hematoxylin-eosin staining.

## Immunofluorescent staining

**Paraffin embedded samples.** Testes were fixed in 0.1M PB-buffered 4% paraformaldehyde (PB-PFA, pH7.4) overnight at 4°C, imbedded in paraffin and sectioned at 6μm. After deparaffinization and rehydration, antigen retrieval was performed by boiling sections in 0.01M sodium citrate (pH 6.0) for 10 min.

**Fresh-frozen samples.** Freshly collected mouse gonads were imbedded in OCT, frozen in liquid nitrogen, and sectioned at 6μm. After fixation in 2% chilled PB-PFA (pH7.4) at 4°C for 10min and PBS washing, antigen retrieval was carried out with PBS containing 1% SDS [48].

**Cultured cell samples.** 24 hours after transfection, 293FT cells were replated on matrigel (BD Biosciences, 354277) coated coverslips. The next day, cells with about 40% confluency were fixed in 2% PB-PFA (pH7.4) for 10min at room temperature, followed by three times PBS washing.

**Meiotic spread samples [49].** After released from tunica albuginea, seminiferous tubules were rinsing in PBS to remove adherent extratubular tissue and then immersed in hypotonic extraction buffer (30mM Tris, 50mM sucrose, 17mM citric acid, 5mM EDTA, 2.5mM DTT, 1mM PMSF, pH 8.2) for 30min. Subsequently, each of one-inch-long seminiferous tubules were torn into pieces with tweezers in 40μL 0.1M sucrose and the result cell suspension was dispersed in 1% PFA containing 0.15% TritonX-100 (pH 9.2) on slides. The slides were kept in a humidified box for one night, air dry in a fume hoods, and washed in 0.04% Photo-Flo (Kodak) for 4min.

**Seminiferous tubule Squash samples [50].** Seminiferous tubules were obtained through above stated procedure, and every 20mm long tubules were fixed in 100μL fix-lysis solution (0.1M PB, 0.8% PFA, 0.1% TritonX-100, pH9) on slide for 10min at room temperature. After removal of the excess solution with filter paper, a coverslip was placed on the slide, and pressure was applied on the coverslip by palm to disperse cells from the tubules. Slide was frozen in liquid nitrogen for 15 seconds immediately. Followed by removing coverslips with a lancet, slides were washed in PBS for three times.

**TUNEL assay.** TUNEL labeling was performed before antibody incubation procedure with a one step TUNEL assay kit (Beyotime, C1089). Specimens were incubated in detection buffer at room temperature for one hour, followed by three times washing in PBS.

**Staining procedure.** Specimens were first blocked in TBS buffer (with 10% normal goat serum, 3% bovine serum albumin) at room temperature for 60min, and incubated with primary antibodies overnight at 4°C. Secondary antibodies incubation was performed at room temperature for 2 hours. 6μg/mL Hoechst33342 (Beyotime, C1022) was used to counterstain cell nuclear. All stains were diluted in blocking buffer, and washing was conducted in TBST between each incubation. Finally, sections were mounted by antifade reagent (Life Technologies, 9071S) and coverslips. Immunostainings were visualized and captured by confocal microscopy (Nikon, A1 RSiMP) or super-resolution structure illumination microscopy (Nikon, SIM-S). Images were further analyzed by NIS-Elements software (Nikon). Information about antibodies were listed in S7 Table.

## Testicular cell transplantation

Testicular cell transplantation was performed according to previously published protocol with slight modifications [51,52]. Briefly, 6 week-age mice were intraperitoneally injected with

40mg/kg Busulfan (Sigma, B2635) to deplete the endogenous germ cells. One month after Busulfan treatment, the mice were used as recipient. For donor cells, 3 dpp-10 dpp seminiferous tubules were dissociated into single cells by trypsin-EDTA, filtered by 70μm cell strainer, and resuspended in ice-cold PBS supplementary with 0.04% Trypan blue at concentration of $1\times10^{8}$ cells/mL. Cells were then loaded into the injection instrument, a glass capillary with 40μm sharpened tip connected to a 1mL syringe by teflon capillary. Immediately after loading, the tip of the glass capillary was inserted through the bundle of efferent ducts and piercing into the rete testis of anesthetized recipient (S4B Fig). By applying gently pressure to the syringe, about 10μL donor cells were filled into the seminiferous tubules of each recipient testis. 4 months after transplantation, testes were collected from the recipient mice for subsequent analysis. At the time of recovery, all the non-SSCs donor cells have been excreted out from the recipient testis as indicated by trypan blue.

## Flow cytometry

Fetal gonads were digested with TrypLE Express (Gibco, 12605010) to obtain single cell suspension. After removing the tunica albuginea, postnatal testes were first digested in collagenase IV (Gibco, 17104019) to remove the interstitial cells, followed by 3 times washing with PBS, seminiferous tubules were further digested with 0.25% trypsin-EDTA (Gibco, 25200056), collagenase IV and DNase I (Solarbio, D8071). After neutralization with fetal bovine serum, cells were washed with PBS and filtered with 70μm strainer. Single cells were finally resuspended in HBSS buffer containing 2% fetal bovine serum and 10mM HEPES. For Hoechst33342 staining assay, seminiferous tubules derived single cell suspensions were stained with 5μg/mL Hoechst33342 for 1 hour at 32˚C as previously described [53,54]. See S4C and S4D Fig for detailed control and gating strategy. Cells were analysis or sorted by Influx (BD Biosciences), and data were analyzed and displayed by FlowJo. Every FACS result was repeated for at least three times with three biological duplicates.

## RNA-seq

Leptotene (9dpp) and pachytene (17dpp) spermatocytes were sorted from VPHS; *Igsf11*$^{-/-}$ or VPHS; *Igsf11*$^{+/-}$ mouse testis, respectively, using the VPHS reporter system. Gatings for FACS were shown in Fig 1E (panels g and i red color). Purity of the sorted cells were analysis by Q-PCR (S6D Fig) and meiotic spread staining (S2C and S2D Fig). Total RNAs were purified with TRIzol reagent (Life Technologies, 15596) from each of $3\times10^{5}$ sorted cells. mRNA libraries were constructed by Smart-seq2 method [55]. Briefly, 80ng of total RNA from each sample was used as input for reverse transcription with SuperScript II reverse transcriptase (Life Technology, 18064), followed by 8 cycle pre-amplification with a hot start PCR kit (KAPA Biosystems, KR0370). After purification with magnetic beads (Beckman Coulter, A63881), 5ng of each amplified cDNA was fragmented with a library preparation kit (Vazyme, TD502) and index (Vazyme, TD202) was added through 8 cycle enrichment PCR according to the manufacturer's protocol. Fragments around 350 bp were selected and all libraries were sequenced by Illumina HiSeqX10.

For RNA-seq data analysis, adaptors were removed with Cutadapt [56], gene expression levels were quantified with Kallisto [57], differentially expressed genes (criteria: P value<0.05 and fold change>2) were identified by DESeq2 [58], and gene ontology was performed with DAVID [59]. Sequencing data is available in the GEO database (GSE174752). The expression pattern of selected DEG genes were further verified by Q-PCR.

## sisHi-C library generation and data processing

Diplotene stage spermatocytes were sorted by using the VPHS reporter system. Purity of the sorted cells were analysis by meiotic spread (S2C and S2D Fig). And sisHi-C libraries were produced as described [60]. Briefly, samples were cross-linked with 1% formaldehyde at room temperature (RT) for 10 min. Formaldehyde was quenched with glycine at RT for 10 min. The cells were lysed on ice for 50min and the chromatin was solubilized with 0.5% SDS. The nuclei were digested with MboI at 37°C overnight. After fill-in with biotin-14-dCTP, the fragments were ligated at RT for 5.5h. These was followed by reversal of cross-linking, DNA purification, and sonication. Then the biotin-labeled DNA was pulled down with 10μl Dynabeads MyOne Streptavidin C1 (Life Technology). The fragments that included a ligation junction were subjected to Illumina library preparation, which including end repair, dATP tailing and adaptor ligation. 12–15 cycles of PCR amplification were performed with TransStart Fastpfu (Transgene), and the products were purified and size-selected with AMPure XP beads. Fragments ranging from 200 bp to 500 bp were selected. All libraries were sequenced on an Illumina HiSeq 1500 according to the manufacturer's instructions.

Paired end Hi-C sequencing reads were processed using HiC-Pro [61]. Briefly, paired reads were iteratively mapped to the mm9 reference genome via bowtie2 (v 2.3.5.1). The unmapped reads, multiple mapped reads and uninformative reads pairs from self-circle ligation, dangling ends, undigested DNA fragments, PCR artifacts were discarded, only the valid read pairs from two different restriction fragments were retained and we merged biological replicates valid pairs for following analysis. The Hi-C matrices were further converted to Hi-C contact matrices binned into 1Mb, 300kb, 100kb sizes, respectively, and normalized by an iterative correction method [61]. Then, Hi-C matrices were transferred into the hic format to facilitate visualization with juicebox [62]. The pairwise differential heatmap were calculated using HiC-Plotter under 300kb resolution of the log2 comparison matrix [63]. For P(s) curves calculation, the contact probability versus genomic distance (P(s)) curves were calculated with normalized interaction matrices in 100kb resolution as previously described in [60,64]. For A/B Compartment analysis, the compartmentalization strength was calculated at 100kb resolution using HiCExplorer [65]. The A/B compartment profile were identified using the eigenvector of the first principal component [66]. The eigenvector of the first principal component defines the A/B compartment profile ('A': active/euchromatic compartments and 'B': inactive/heterochromatic compartments) primarily based on mRNA transcription. For Hi-C biological replicate correlation analysis, the Peason correlation between sisHi-C biological replicates was calculated for each pair of 100kb bins within a maximum distance of 50 bins (5Mb) as previously described [67]. Sequencing data is available in the GEO database (GSE174752).

## Statistics

Values are given as mean ± S.D. (Standard Deviation) and statistical analysis was performed with two-tail Student's t-tests.

## Supporting information

**S1 Fig. Characterization of the VPHS meiotic reporter mice. (A)** Design of guide RNAs for *Sycp3* targeting. Guide RNA recognition sequences and protospacer-adjacent motif (PAM) are shown on yellow or green background, Cas9 cleavage sites were indicated by triangular arrowheads. The downstream cleavage site is 51 bases upstream the transcription start site (TSS) of *Sycp3*. Indels that caused by Cas9-mediated double strand breaks in targeted mouse embryos as confirmed by sequencing. Abbreviations (the same as below): D10, Deletion 10 base. **(B)** Histological analysis of VPHS transgenic testes. Arrowheads indicate degenerated cells

occasionally found in VPHS homozygote. **(C)** Expression of total *Sycp3* transcript (ALL *Sycp3*) in adult testis of different VPHS genotypes. Biological repeats = 3, Technical repeats = 3. **(D)** Expression of wild type (WTS) and knockin (KIS) *Sycp3* transcript in adult testis with different VPHS genotypes. Biological repeats = 3, Technical repeats = 3. **(E-H)** Protein expression of VPHS transgene in adult testis (E-G). Triangle star indicates the region if non-cleavage mVenus-P2A-HA-SYCP3 (59kD) appear. Relative SYCP3 levels were normalized to UBC9 (H). n = 2 animals/genotype. **(I)** Detection of transgenic HA-SYCP3 in 17 dpp seminiferous tubules of different VPHS genotypes. Values and error bars are mean and SD. *, P < 0.05; **, P < 0.01 by Student's t test. ND, not detected.
(TIF)

**S2 Fig. Characterization of the VPHS meiotic reporter mice. (A)** Comparison of mVenus fluorescent intensity between 30dpp VPHS homozygote (Homo) and heterozygote (Hetero) littermates. **(B)** Comparison of mVenus fluorescent intensity in homozygote gonads between each developmental stage. Separated diagram for each developmental stage was shown in Fig 1E. **(C-D)** Meiotic spread staining analysis (C) and statistics (D) of spermatocytes sorted by the VPHS reporter from different developmental stages. Cells were sorted by the red gating shown in Fig 1E. n = 899 cells. **(E)** Detection of mVenus fluorescence within homozygotes fetal testis of the VPHS mice. **(F)** Expression of transgenic *Sycp3* transcript within embryonic homozygote VPHS gonads. For each development stage, fetal gonads from two pregnant mice were pooled as one biological repeat for RNA extraction. Biological repeats = 1, Technical repeats = 3. Values and error bars are means and S.D. **, P < 0.01 by Student's t test.
(TIF)

**S3 Fig. Expression of *Igsf11* within mouse testis. (A)** Genotyping of *Igsf11* knockout mice by melting curve of the Q-PCR product. **(B)** Illustration of IGSF11 as transmembrane protein. **(C)** Expression of *Igsf11* in different stage of fetal gonad. Biological repeats = 1, Technical repeats = 2. Values and error bars are mean and SD. **(D)** Purification of mouse Sertoli cells by FACS. **(E)** Single cell expression profiling of *Igsf11* in postnatal spermatogenic cells from previous study [20]. Abbreviations: A1S, type A1 spermatogonia; TBS, type B spermatogonia; LEP, leptotene spermatocyte; ZYG, zygotene spermatocyte, PAC, pachytene spermatocyte; DIP, diplotene spermatocyte; DIV, metaphase spermatocyte; RS2, steps 1–2 spermatids; RS4, steps 3–4 spermatids; RS6, steps 5–6 spermatids; RS8, steps 7–8 spermatids. n, number of spermatocytes. n = number of single cells. **(F-G)** Specificity of the anti-IGSF11 antibody was confirmed in human 293FT cells expressing HA-IGSF11 by Western blot (F) or immunostaining (G). n = 3 experiments. Abbreviations: NC, negative control; OE, overexpression IGSF11.
(TIF)

**S4 Fig. Analyze the development status of testicular cells by FACS. (A)** Immunostaining of HA-IGSF11 in 293FT cells. Images were captured by confocal or super resolution SIM microscope. **(B)** Example of testicular cell transplantation surgery. Trypsinized testicular cells were resuspended in PBS containing Trypan Blue and injected into the recipient testis. Trypan Blue indicated donor cell-filled seminiferous tubules. **(C)** Gating strategy of V&H assay using VPHS transgenic testis. **(D)** Controls that help for the gating of VPHS+ cells. **(D)** Comparison between the results of V&H assay and traditional Hoechst33342 staining assay.
(TIF)

**S5 Fig. Formation and structure of BTB is not affected in the absence of *Igsf11*. (A)** Immunostaining of ZO-1 and CX43 in control and *Igsf11* knockout adult testis. Abbreviations: ob, old BTB. **(B)** Transmission electron microscopy analysis of control and *Igsf11* knockout 17dpp

testis. BTB was indicated by arrow heads.
(TIF)

**S6 Fig. *Igsf11* deficient mouse spermatocytes develop beyond pachytene checkpoint. (A)** Comparison of the axial element assembly in spermatocytes of the control (*Igsf11*[+/-]) and mutant (*Igsf11*[-/-]) mice using SYCP3 and REC8 as markers. **(B)** Meiotic progression analysis of the first wave spermatocytes with the VPHS transgenic reporter. **(C)** Q-PCR quantification of typical retrotransposon expression in purified leptotene and pachytene spermatocytes with different *Igsf11* genotypes. Biological repeats = 3, Technical repeats = 3. **(D-G)** Transcriptome comparison of leptotene and pachytene spermatocytes between *Igsf11* genotypes (E). Biological repeats = 2. Leptotene and pachytene spermatocytes were sorted with the help of the VPHS reporter (Fig 1E). Purity of the sorted cells were analysis by Q-PCR (D) and meiotic spread staining (S2C and S2D Fig). Gene ontology of up-regulated genes in *Igsf11*[-/-] pachytene spermatocytes (F). The expression of sex chromosome-linked genes (*Cdc25c*, *Hprt*, *Mecp2*, *Pgk1*, *Rbmy1a1*, *Ube1y1*) and compensatory genes on autosomes (*Cetn1*, *Pgk2*) that extracted from above mRNA-seq data (G). Abbreviations: lep, leptotene; pac, pachytene. **(H)** Incorporation analysis of H1t within meiotic prophase I spermatocytes.
(TIF)

**S7 Fig. *Igsf11* deficiency leads to nonhomologous pericentric heterochromatin clustering in mouse primary spermatocytes. (A)** Organization dynamic of PCH within primary spermatocytes of different *Igsf11* genotypes. **(B)** Statistics of SYCP3-positive cells after early-pachytene with meiotic spread specimens. The normal or defective developmental status of the SC or PCH is indicated by "√" or "×", respectively. n = 868 spermatocytes. **(C-E)** Comparison of compartmentalization (C) and compartmentalization strength (D) between wild type and *Igsf11* knockout spermatocytes. **(E)** Correlation of normalized interaction frequency among biological replicates in the Hi-C data.
(TIF)

**S1 Table. Statistics of testicular cell transplantation experiment.**
(XLSX)

**S2 Table. Distribution statistics of meiotic prophase I cells in adult testes of different *Igsf11* genotypes.**
(XLSX)

**S3 Table. RNA-seq analysis of purified spermatocytes.**
(XLSX)

**S4 Table. Current germ cell reporter in mouse.**
(XLSX)

**S5 Table. Genotyping protocol for VPHS mice and *Igsf11* knockout mice.**
(XLSX)

**S6 Table. Primers for Q-PCR.**
(XLSX)

**S7 Table. Primary and secondary antibodies.**
(XLSX)

**S1 Appendix. DNA sequence of recombinant *Sycp3* exon 1 in VPHS mice.**
(DOCX)

## Acknowledgments

We gratefully acknowledge Prof. Fei Gao (Institute of Zoology, Chinese Academy of Sciences), Prof. Jie Na (School of Medicine, Tsinghua University) for sharing transgenic mice in this study; Prof. Scott Keeney (Sloan Kettering Institute, Memorial Sloan Kettering Cancer Center), Prof. Qingyuan Sun (Institute of Zoology, Chinese Academy of Sciences), Prof. Wei Xie (School of Life Sciences, Tsinghua University) for sharing antibodies and Prof. Scott Keeney for commenting on our manuscript; Yan Liu, Jinyu Wang and Huizhen Cao for technical assistance.

## Author Contributions

**Conceptualization:** Bo Chen, Kehkooi Kee.

**Data curation:** Bo Chen.

**Formal analysis:** Bo Chen, Jing He, Yang Liu, Xuerui Yang.

**Funding acquisition:** Kehkooi Kee.

**Investigation:** Bo Chen, An Yan, Lin Li.

**Methodology:** Bo Chen.

**Project administration:** Kehkooi Kee.

**Resources:** Gengzhen Zhu, Chen Dong.

**Supervision:** Kehkooi Kee.

**Writing – original draft:** Bo Chen, An Yan, Jing He.

**Writing – review & editing:** Bo Chen, Kehkooi Kee.

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
