## [Decision Letter · Decision Letter 0]

29 Jun 2021

Dear Dr Kee,

Thank you for submitting your manuscript entitled: "IGSF11 is required for pericentric heterochromatin dissociation during meiotic diplotene" to PloS Genetics. I am enclosing the comments that reviewers made on your paper (and I'm really sorry for the unusually long review process and large number of referees, but all accepted to review!). Your manuscript has now been seen by our 5 referees, whose comments are appended below. In the light of their advice we would be glad to consider a revised version of the paper addressing the reviewer concerns. As you will see most comments are quite easy to address. There are also some comments about the conclusion, which may need to be more carefully considered (Reviewer 1). I hope you will find the reviewers' reports useful and constructive. Please don't hesitate to get in touch if you have any questions regarding the revisions or would like to discuss a revision plan.

[LINK]

Yours sincerely,

Sue Hammoud, Ph.D.

Guest Editor

PLOS Genetics

Gregory P. Copenhaver

Editor-in-Chief

PLOS Genetics

Reviewer's Responses to Questions

**Comments to the Authors:**

Reviewer #1: In meiosis I, chromosomes exhibit a unique segregation pattern as homologous pairs segregate to different poles of the dividing cell. Studies such this on mechanisms that guarantee the proper segregation of chromosomes are important because errors in meiosis I are important causes of birth defects and mental retardation.

Proper meiotic segregation patterns are achieved by meiosis-specific mechanisms such recombination, dissolution of the synaptonemal complex, and other aspects that alter chromosome structure and behavior such homologous and heterologous centromeric interactions. In this work by Chen et. al. the authors created a mouse model based on the expression of GFP under the control of the Sycp3 gene promoter and use this model for phenotypic analysis of IGSF11 knockout spermatocytes and oocytes.

A remarkable aspect of this work is the generation and validation of Venus-P2A-HA-Sycp3. This alone may constitute a significative addition to the much-needed toolbox of mouse models and molecular strategies in the field. Except for relative minor observations (detailed next), the idea of a mice that can be used to easily isolate meiocytes at different stages of prophase I is exiting, and the data validating the model in this work is of excellent quality.

Observations regarding this part of the work: Page 6. VPHS mice seems to have smaller testis. This together with a described effect of the described knock-in in specific variants expression of Sycp3, bring to my attention that this possible deficiency may have consequences in gamete development. Here, a careful characterization should determine differences of VPHS testis respect to control. This is by measuring testis weight, typifying seminiferous tubule types, and addressing apoptosis, among other possible analysis. In this endeavor, it may help clarifying what is known regarding the possible functions of a smaller Sycp3 variant. Although this should be addressed, the fact that DNA repair and early and mid-stages of meiosis are apparently normal is very encouraging.

In a second aspect, this manuscript describes the phenotype of IGSF11 deletion and claim that in absence of this protein cells arrest at diplotene like stage and non-homologous pericentromeric heterochromatin dissociation is affected. As presented the evidence may not support the claims and they need to strengthen the presentation of data.

1- Page 15, starting in line 309. “Spermatocytes cannot develop beyond diplotene”. Evaluation of figures leads me to the conclusion that abnormal diplotene may actually represent apoptotic pachytene-like cells? In this case, the manuscript will need adjustments in the interpretation of the data.

In the same stream, it is possible that the process of exit from pachytene to be affected, accounting for other described phenotypes. A possible way to address this is by evaluating the state of chaperones such as HSP70 and partners. Related to this, in Figure 6 and page 18, again, are the “diplotene” cells actually in diplotene stage, or they are apoptotic cells? So actually, that results may be explained by cells being arrested in pachytene and accumulating defects as they become apoptotic?

Also, I think the focus of attention should be late pachytene and early diplotene (please quantitate), where axis decomposition has not yet occurred.

2- It is possible that failures observed in IGSF11 knockouts could be ascribed to changes in chromosome associations to the nuclear membrane (perhaps related to the LINC complex?) This could explain changes in pericentromeric chromatin clusters and relative later prophase I phenotype (i.e., increased chromosome entanglements?). This view is supported by IGSF11 being a membrane protein. In principle, this could be assessed by evaluation of telomere and LINC proteins localization in spermatocyte squashes.

Other comments (not in a specific order):

1- The work fails to describe previous relevant work on pericentromeric chromatin association dynamics in mice (e.g., work from the Dawson and Sherthan labs). A careful comparative analysis will be useful for readers and to determine significance of this work.

2- Excess of acronyms are complicating the reading, for example HT and HO.

3- Figure 1E. Indicate % corresponding to Leptotene and zygotene in 20 dpp and if possible, in Figure S2G.

4- Methods and Materials description is insufficient. Please, revise the entire Method section with special attention to a- RNA-seq methods and analysis of results, and 2-obtention and validation of enriched fractions of meiotic and not meiotic cell populations. Validation of the enriched cell fractions is very important to assess the accuracy of results interpretation.

5- Quantitation and reproducibility:

a- In all experiments is not immediately clear number of mice used and biological replicates. Please revise and add in the text or figure legend. In the same line, please review and reach minimum statistical number (or at least justify current data validation) for transplant experiments and cell sorting experiments.

b- Quantitation (or mentioning the number of repetitions) may be required in several figures and experiments. For example, Fig 1F, Fig 2H, Fig 4, Fig 5, and FigS5B, part of Fig 6.

c- RNA seq experiments are inadequately described leaving critical information regarding procedure and statistics (e.g., repetition) incomplete. Equally important, the description of results in the text is not sufficient. As presented data do not support conclusions in page 17 line, 347-352. Careful analysis of gene sets for x-y inactivation, retrotransposon silencing, and genome wide transcription changes (particularly in genes that may affect synapsis, recombination repair and other essential meiotic process) need to be better assessed and documented. Why do they use sex chromosome linked genes? Is this to assess X-Y silencing? Are genes involved in pachytene exit affected (these are not part of the pachytene checkpoint). Regarding the material used, what purified leptotene and pachytene means?

d- They should also consider that even if transcription is not affected, localization of critical factor may, thus causing phenotype.

e- Quantitation of experiments appear in supplementary data, please integrate this data into the main figure (example, testis size quantitation and data in FigS6A).

f- please provide entire data sets.

6- Page 13, line 276. Replace significantly for apparently.

7- Fig 4B – page 14, line 290. What criteria was used to classify Pre-leptotene cells. This is not immediately clear as no marker was used.

8- In figures and text (example pag15, line 316). What abnormal diplotene means? What is the criteria to classify these cells as diplotene? As indicated above, are they diplotene or cells arrested in pachytene and undergoing apoptosis?

Reviewer #2: How germ cells and gonadal somatic cells work together to regulate meiosis is unclear. In this manuscript, the authors constructed a mVenus-P2A-HA-Sycp3 reporter mice which can be used to examine the progression of meiotic prophase I and isolate meiotic cells from mouse testes. Then, the authors identified an adhesion molecule IGSF11, which is expressed in both sertoli cells and germ cells. With the help of the new reporter mice and testicular cell transplantation, they demonstrated that IGSF11 expressed in both sertoli and germ cells is required for meiosis completion in spermatocytes. The authors further showed that in the absence of IGSF11, spermatocytes proceed through pachytene without obvious defects, but the pericentric heterochromatin of nonhomologous chromosomes could not dissociate properly. The abnormal interchromosomal interactions was further confirmed by Hi-C analysis. This study suggests IGSF11 plays an important role in regulating the coordination between sertoli and germ cells for proper meiosis progression. Experiments are well done and convincing. I only have several minor concerns.

1. Page6 line 126, please specify "corresponding expression levels".

2. p7 lies130-135, does the longer isoform have any special functions?

3. P7, it would be better to show the reporter mice have normal fractions of spermatocytes in different substages (leptotene, zygotene, etc).

4. P10 line 203, authors should note that SYCP3 is an axis component.

5. Figure 1 and the legends: I did not see gating parameters in Figure 1E.

Is it possible to do the same analysis as in panel F to confirm the sorted meiotic cells in female?

6. Figure 2C, IGSF11 have two isoforms? If so, please clarify, if not, please explain whay there were two bands.

7. Figure 2FG, please specify the error bar and how many times experiments were repeated.

Reviewer #3: In this interesting manuscript the authors report on the male infertility phenotype of deficiency of the immunoglobulin-like cell adhesion molecule IGSF11, significantly and informatively adding to a previous study. They document IGFS11 expression in both Sertoli cells and germ cells and show that targeted mutation of the Igfs11 gene leads to infertility with arrest of spermatocytes at a late pachytene stage, prior to diplotene. Germ-cell transplantation analyses revealed that for normal gametogenesis, IGSF11 is required in both Sertoli cells and germ cells. The authors show abnormal clustering of the pericentric heterochromatin (PCH) of nonhomologous chromosomes at late pachytene, shown also by Hi-C analysis, resulting in failure of chromosome individualization. Another nice feature of this manuscript, albeit somewhat unrelated to the main theme, is the development of a mVenus SYCP3 reporter system (VPHS) that facilitated analysis of the progress of meiosis in spermatocytes as well as cellular localization in testis tubules.

Overall, this is a great scientific study, presenting new information, and commendable for the depth of analysis and incredible thoroughness in attention to detail. The work is competently performed, with appropriate controls, and the figures are, for the most part, excellent, with only a few suggestions for improvement.

The transplantation experiments are especially appreciated. However, this should have a bit more explanatory rationale for readers unfamiliar with the technique.

The major concern about this manuscript is with respect to the striking similarity of the Igsf11 KO germ-cell phenotype to the germ-cell phenotype in mice lacking androgen receptor in Sertoli cells (the SCARKO mutant model). Clearly this study has provided more information about the abnormal diplotene and compromised dissociation of the PCH, which has not been explicitly investigated in SCARKO mice. However, it is difficult to imagine a direct mechanism by which IGSF11 would control PCH dissociation, rather than it being a consequence of arrest of meiotic progression. Thus, it would be interesting for the authors should comment on the similarities and differences of Igsf11 KO and SCARKO phenotypes; this consideration could yield insights into possible mechanisms of IGSF11 action (could it be downstream of AR, and facilitate androgen signaling to the germ cells?). It would also be informative to comment on the apparent lack of differences in the genome-wide transcription profile of the Igsf11 mutants in the context to what is known about gene expression in SCARKO mutants.

Other editorial comments:

1. The transgenic VPHS reporter will be a useful resource to others; information on how it will be made available would be helpful.

2. The authors should take care not to confuse DNA content, C, with chromosome content, N. For example, spermatocytes are 4C (post S-phase), but 2C (and definitely not “tetraploid,” as often erroneously stated in the literature!). This must be corrected throughout the text and figures.

3. Fig. 2H needs better legend and labels to point out features discussed in text, as well as clearer validation of the specificity of the antibody. As presented, it appears that staining is not robust (Fig. 2l-n) or histology is unclear (Fig. 2o-r).

4. Fig. 4B (“ob” and “nb”) is difficult to interpret and could be improved with either (or both) higher magnification or a diagram. Alternatively, Fig. S5B could be pulled into the main text as more convincingly illustrative of the BTB.

Reviewer #4: The authors developed an ingenious meiotic reporter based on the knock-in insertion of a fluorescent protein in the meiotic gene Sycp3. They combined this reporter with an pre-existing Hoechst staining-based sorting protocol in order to sort meiotic germ cells at different stages among testes cells with increased specificity. They then generated a KO mouse model for IGSF11, an adhesion protein expressed both in Sertoli cells and in germ cells within the testis. IGSF11 is required for completing male meiotic prophase, and the authors show by an elegant testis cell transplantation assay that IGSF11 expression is required both in Sertoli cells and in germ cells. They show that spermatogenesis is arrested during the diplotene stage of meiotic prophase I, representing a new arrest point, and that it is characterized by a defect in chromosome reorganization within the nucleus in the late prophase. Although this report does not permit to understand the mechanism by which the lack of the adhesion protein IGSF11 leads to this defect, there is no doubt that it provides an insightful description of the spermatogenesis defects of this mutant that will be a reference for further studies (in addition to describing the generation of a novel, useful reporter). The study is conducted well, the experiments are well done and interpreted carefully, and the manuscript is clear and well written. I have only few concerns listed below.

Pages 15-17: The abnormal diplotene stage shown on figures 5 and S6 is not described in the main text. Early, middle and late-diplotene are shown on figure 6A, but it is not explained how these substages are defined in the mutant, where SYCP3-defined axes appear to become fragmented faster than in WT. What is the proportion of each sub-stage compared to WT? This would give information about whether the spermatocytes are arrested and/or eliminated throughout diplotene stage or at the end of it. A more detailed analysis of histological sections would allow to define more precisely the stage of spermatocyte apoptosis (e.g. stage XII, or earlier).

Fig 6E and 7: HiC data shown on figure 7D indicate a relative increase of end-end interchromosomal interactions, including proximal-distal interactions. This is reminiscent of the bouquet stage, which occurs normally earlier and is characterized by a reduced number of PCH clusters, too. This might be examined with an additional centromere-specific marker, which would allow distinguishing proximal from distal telomeres on spermatocyte spreads. It is indeed not clear whether only proximal telomeres associate with the clusters of PCH, nor whether every proximal telomere does so.

Discussion: A disfunction of BTB was suggested as a possible main cause for infertility in ref 8. The authors propose here a maybe more subtle defect because of the normal-looking BTB. This discrepancy should be discussed. More generally, whereas the germ cell arrest is often reported to occur much earlier than diplotene stage in mutants disrupting BTB function, this is not always clear and should be discussed. While I’m not a specialist, I’m not sure that the histological examination (markers and EM) is sufficient to conclude about the integrity of the BTB without a functional assay (diffusion of some compound such as shown in Xu et al., 2009, Mol Biol Cell 20:4268).

Reviewer #5: see attachment

**Have all data underlying the figures and results presented in the manuscript been provided?**

Reviewer #1: Yes

Reviewer #2: Yes

Reviewer #3: Yes

Reviewer #4: Yes

Reviewer #5: Yes

PLOS authors have the option to publish the peer review history of their article (what does this mean?). If published, this will include your full peer review and any attached files.

Reviewer #1: No

Reviewer #2: **Yes: **liangran zhang

Reviewer #3: No

Reviewer #4: No

Reviewer #5: No

---

## [Editor Report · Decision Letter 1]

16 Aug 2021

Dear Dr Kee

We are pleased to inform you that your manuscript entitled "IGSF11 is required for pericentric heterochromatin dissociation during meiotic diplotene" has been editorially accepted for publication in PLOS Genetics. Congratulations!

Yours sincerely,

Sue Hammoud, Ph.D.

Guest Editor

PLOS Genetics

Gregory P. Copenhaver

Editor-in-Chief

PLOS Genetics

Comments from the editors:

Thank you again for your submission to PLOS Genetics. It is my pleasure to write and let you know that your manuscript, "IGSF11 is required for pericentric heterochromatin dissociation during meiotic diplotene" is now acceptable for publication.

**Data Deposition**

http://datadryad.org/submit?journalID=pgenetics&manu=PGENETICS-D-21-00672R1

**Press Queries**

---

## [Editor Report · Acceptance letter]

2 Sep 2021

PGENETICS-D-21-00672R1 

IGSF11 is required for pericentric heterochromatin dissociation during meiotic diplotene 

Dear Dr Kee, 

We are pleased to inform you that your manuscript entitled "IGSF11 is required for pericentric heterochromatin dissociation during meiotic diplotene" has been formally accepted for publication in PLOS Genetics! Your manuscript is now with our production department and you will be notified of the publication date in due course.

With kind regards,

Katalin Szabo

PLOS Genetics

On behalf of:
